# Uniform Mean Estimation for Heavy-Tailed Distributions via Median-of-Means

**Mikael Møller Høgsgaard** [* 1]   **Andrea Paudice** [* 1]

## Abstract

The Median of Means (MoM) is a mean estimator that has gained popularity in the context of heavy-tailed data. In this work, we analyze its performance in the task of simultaneously estimating the mean of each function in a class $\mathcal{F}$ when the data distribution possesses only the first $p$ moments for $p \in (1, 2]$. We prove a new sample complexity bound using a novel symmetrization technique that may be of independent interest. Additionally, we present applications of our result to $k$-means clustering with unbounded inputs and linear regression with general losses, improving upon existing works.

## 1. Introduction

The problem of estimating the mean of a random variable from a finite sample of its i.i.d. copies is fundamental in statistics and machine learning. When the random variable has exponentially decaying tails, the sample mean exhibits optimal or near-optimal performance. In particular, for $\varepsilon, \delta \in (0, 1)$, it is known that $\text{POLYLOG}(1/\delta)/\varepsilon^2$ samples suffice to obtain an $\varepsilon$-close estimate with probability at least $1 - \delta$. Recent studies have shown that heavier-tailed distributions, possessing only the first $p$ moments for $p \in (1, 2]$, are better suited to model several important cases, including but not limited to, large attention and language models (Zhang et al., 2020; Zhou et al., 2020; Gurbuzbalaban et al., 2021; Gurbuzbalaban & Hu, 2021), certain applications in econometrics (Bradley & Taqqu, 2003) and network science (Barabási, 2016), and some classes of extremal processes (Nair et al., 2022). Under this model, the sample mean suffers from sub-optimal performance with a polynomial dependence on $1/\delta$ (Catoni, 2012). *Median-of-Means* (MoM) is a mean estimator that provides optimal performance guarantees even under heavy-tailed distributions (Nemirovskij

& Yudin, 1983; Jerrum et al., 1986; Alon et al., 1996). Its popularity is largely due to its simplicity and efficiency. Indeed, its computation only requires splitting the sample into $\kappa$ batches, computing the sample mean in each batch, and then returning the median of these sample means, with an overall runtime that is quasi-linear in the number of observations. Notice that the user is only required to specify the number of batches, which should be of order $\log(1/\delta)$ for optimal performance.

In this work, we analyze the performance of the MoM estimator in solving the following significant generalization of the mean estimation task, a problem typically referred to as *uniform convergence*. Given a set of real-valued functions $\mathcal{F}$ over a *domain* $\mathcal{X}$, and a distribution $\mathcal{D}$ supported over $\mathcal{X}$, we consider the problem of estimating, simultaneously for each $f \in \mathcal{F}$, the mean $\mu(f) = \mathbb{E}[f(\mathbf{X})]$ from an i.i.d. sample $\mathbf{X} \sim \mathcal{D}^n$ generated from $\mathcal{D}$. In particular, our goal is to estimate the *sample complexity* of the MoM estimator, i.e., the smallest sample size $n^* = n(\varepsilon, \delta, \mathcal{F})$ that suffices to guarantee that for all $\varepsilon, \delta \in (0, 1)$ and $n \geq n^*$, the following holds:

$$\mathbb{P}_{\mathbf{X} \sim \mathcal{D}^n} \left( \sup_{f \in \mathcal{F}} | \text{MOM}(f, \mathbf{X}) - \mu(f)| \leq \varepsilon \right) \geq 1 - \delta. \quad (1)$$

Uniform convergence has fundamental applications in machine learning. First, given an estimator $\theta$ satisfying (1), one can *learn* $\mathcal{F}$ by minimizing $\theta(f, \mathbf{X})$ over $\mathcal{F}$. Notice that if $\theta$ is the sample mean, this corresponds to the standard *Empirical Risk Minimization* (ERM) paradigm. Second, such an estimator can be used to estimate the risk of any function in $\mathcal{F}$ using the same data as for training. This is particularly useful when a test set cannot be set aside, or only an approximate solution to the empirical problem can be computed. Third, as the sample complexity of $\theta$ features a dependence on some complexity measure of $\mathcal{F}$, it can be used to perform *model selection*, i.e., to select a class of functions for the learning problem at hand before having a look at the data.

**Contributions.** We provide the following contributions.

- We show that, upon $\mathcal{F}$ admitting a *suitable* distribution-dependent approximation of size $N_{\mathcal{D}}(\varepsilon, (v_p/\varepsilon^p)^{1/(p-1)})$, where $v_p$ is a uniform upper

---

[*]Equal contribution [1]Department of Computer Science, Aarhus University, Denmark. Correspondence to: Mikael Møller Høgsgaard <hogsgaard@cs.au.dk>, Andrea Paudice <apaudice@cs.au.dk>.

*Proceedings of the 42$^{nd}$ International Conference on Machine Learning*, Vancouver, Canada. PMLR 267, 2025. Copyright 2025 by the author(s).

bound to the $L_p$ norm of the functions in $\mathcal{F}$, the sample complexity of the MoM estimator is at most of order $(v_p/\varepsilon^p)^{1/(p-1)} \log(N_{\mathcal{D}}(\varepsilon, (v_p/\varepsilon^p)^{1/(p-1)})/\delta)$. Specifically, we require that: given $\varepsilon, \delta > 0$ and $m \in \mathbb{N}$, there exists a finite set $F_{(\varepsilon, m)}$ of size at most $N_{\mathcal{D}}(\varepsilon, m)$ s.t. for a large enough $\kappa$, with probability at least $1 - \delta$ the functions in $\mathcal{F}$ can be $\varepsilon$-approximated on most of the $\kappa$ batches of 3 i.i.d. random samples $\mathbf{X}_0, \mathbf{X}_1, \mathbf{X}_2$ of size $m \cdot \kappa$. We argue that this condition on $\mathcal{F}$ is mild, and in addition to capture the canonical case of functions with bounded range, it also captures important classes of unbounded functions.

- To illustrate this we show that our result applies to two important class of unbounded functions. First, we prove a novel *relative* generalization error bound for the classical $k$-means problem that, compared with prior work, features an exponential improvement in the confidence term $1/\delta$. Second, we use the MoM estimator to derive sample complexity bounds for a large class of regression problems. Our sample complexity bound only requires *continuity* of the loss function along with a bound on the norm of the weight vectors. We also provide a more refined bound in the more specific case of Lipschitz losses. Moreover, our sample complexity bounds match the known results for exponentially tailed distributions, only assuming the existence of the $p$-th moments for $p \in (1, 2]$.

- To derive the main result, we introduce a novel symmetrization technique based on the introduction of an additional *ghost sample*, compared to the standard approach using only one ghost sample. While the first ghost sample is used to symmetrize the mean, the second ghost sample is used to symmetrize the MoM. Analyzing two ghost samples simultaneously requires non-trivial modifications to the canonical discretization and permutation steps. The new discretization step allows for relaxing a uniform approximation over the functions to an approximation at the sample mean level, only requiring most of the sample means to be approximated, which is a desirable feature when dealing with unbounded functions and heavy tailed data.

## 2. Related Work

The study of uniform convergence for classes of real-valued functions is a fundamental topic in statistical learning theory. In the special case of binary-valued functions, a complete (worst-case) characterization is provided by the Vapnik-Chervonenkis dimension of the class (Vapnik & Chervonenkis, 1971). When the range of the functions in $\mathcal{F}$ is bounded within an interval, the problem is known to be solved by the sample mean as soon as the fat-shattering dimension (Kearns & Schapire, 1994) of $\mathcal{F}$ is finite at all

scales (Alon et al., 1997; Bartlett et al., 1996; Colomboni et al., 2025). In particular, the best known upper bounds on the sample complexity of the sample mean are of the order of $\varepsilon^{-2}(\text{fat}_\varepsilon + \log(1/\delta))$, where $\text{fat}_\varepsilon$ denotes the fat-shattering dimension of $\mathcal{F}$ at scale $\varepsilon$.

The variant of the uniform convergence problem considered in this work is a special case of the formulation given in (Oliveira & Resende, 2023) except we don't consider adversarial contaminations. Differently from our work, the authors in (Oliveira & Resende, 2023) analyzed the performance of the *trimmed mean* with a focus on the *estimation error*. Their bounds feature a dependence on a quantity related to *Rademacher complexity* (Bartlett & Mendelson, 2003). Similar results, but in the more restrictive case of $p \in (2, 3]$, have also been obtained by (Minsker, 2019), who considered a different class of estimators interpolating between the Catoni´s estimator (Catoni, 2012) and the MoM. The estimation error of the MoM has been studied in (Lugosi & Mendelson, 2019; Lecué et al., 2020) for $p = 2$. These works also feature a dependence on a quantity related to *Rademacher complexity* of $\mathcal{F}$. Compared to this line of work focussing on the estimation error, our focus is on the sample complexity and is thus more aligned with the results discussed earlier in this section of (Alon et al., 1997; Bartlett et al., 1996; Colomboni et al., 2025). We notice that the Rademacher Complexity depends on the sample size, and thus it is sometimes problematic to derive an explicit sample complexity bound from an estimation error bound. Taking a sample complexity perspective allows for coping with function classes that are otherwise difficult to handle through the Rademacher Complexity such as $k$-means clustering with unbounded input and center spaces, and linear regression with general continuous losses. In that respect, we see our results for $p = 2$ as a complement to these works. We remark that our proof technique differs from the bounded difference arguments proposed in (Lugosi & Mendelson, 2019; Lecué et al., 2020), and instead is based on a novel symmetrization argument that we believe may be of independent interest. In contrast, while (Oliveira & Resende, 2023; Minsker, 2019; Lecué et al., 2020) consider both heavy-tailed distributions and adversarial contaminations, in this work, we focus exclusively on heavy-tailed distributions.

## 3. Sample Complexity Bound

In this section, we describe our main result and provide a sketch of its proofs (we refer to Appendix B for the details).

### 3.1. Notation

We will use boldface letters for random variables and non-boldface letters otherwise. Throughout the section, $\mathbf{b} \sim \{0, 1\}$ will always denote the random variable de-

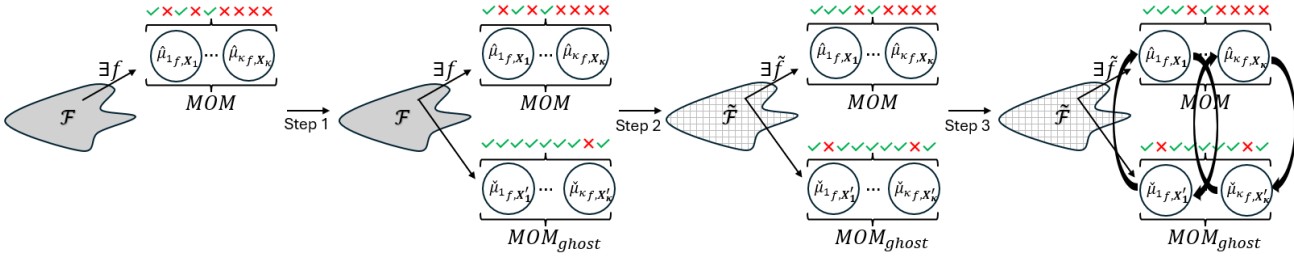

*Figure 1.* Proposed symmetrization approach. Red crosses and green ticks denote mean estimates that failed or succeeded respectively. Step 1: Symmetrization of the MoM with a ghost sample. Step 2: Imbalance preserving discretization of the class $\mathcal{F}$. Step 3: Permutation of the sample means between the MoM of interest and the "ghost" MoM.

fined as $\mathbb{P}_{\mathbf{b}}(\mathbf{b}=0) = \mathbb{P}_{\mathbf{b}}(\mathbf{b}=1) = 1/2$. For a natural number $\kappa \in \mathbb{N}$ we define the set $[\kappa] = \{1,\ldots,\kappa\}$. Given two sets $A$ and $B$, $B^A$ denotes the set of all functions from $A$ to $B$. For a function $f \in \mathcal{F} \subseteq \mathbb{R}^{\mathcal{X}}$, $m \in \mathbb{N}$, $X \in \mathcal{X}^m$, and a distribution $\mathcal{D}$ over $\mathcal{X}$, the notations $\mu_{(f,X)}$ and $\mu_f$ denote the sample mean of $f$ on $X$, i.e. $\mu_{(f,X)} = \sum_{i=1}^m f(X_i)/m$ and its expectation over $\mathcal{D}$ $\mu_f = \mathbb{E}_{\mathbf{X} \sim \mathcal{D}}[f(\mathbf{X})]$. Furthermore, for $p \in (1,2]$ we write $\mathcal{F} \subseteq L_p(\mathcal{D})$ iff $\sup_{f \in \mathcal{F}} \mathbb{E}_{\mathbf{X} \sim \mathcal{D}}[f(\mathbf{X})^p] < \infty$.

For $\kappa \in \mathbb{N}$, if $a_1,\ldots,a_\kappa \in \mathbb{R}$ and we let $a_{(1)} \leq \ldots, \leq a_{(\kappa)}$ denote the numbers in ascending order, we define their median as

$$\text{MEDIAN}(a_1,\ldots,a_\kappa) = \begin{cases} a_{((\kappa+1)/2)} & \text{if } \kappa \text{ is odd} \\ a_{(\kappa/2)} & \text{if } \kappa \text{ is even.} \end{cases}$$

With the definition of the median, we can now define the MoM estimator.

---
**Algorithm 1** Median of Means (MoM) Estimator
---
Input: Sample $X = (X^1,\ldots,X^\kappa)$, $X^i \in \mathcal{X}^m$ $m, \kappa \in \mathbb{N}$, function $f : \mathcal{X} \to \mathbb{R}$
Return: $\text{MOM}(f,X) = \text{MEDIAN}(\mu_{f,X^1},\ldots,\mu_{f,X^\kappa})$.
---

In words, the MOM takes as input a sample consiting of $\kappa$ blocks of $m$ samples in each, and a function $f$ wanting the mean estimate of.

Finally, for $m, \kappa \in \mathbb{N}$, and $X_1, X_2, X_3 \in (\mathcal{X}^m)^\kappa$, for each $l \in \{1,2,3\}$ we rely on the following notation,

$$X_l = (X_{l,1}^1,\ldots,X_{l,m}^1,\ldots,X_{l,1}^\kappa,\ldots,X_{l,m}^\kappa).$$

### 3.2. Proof Overview

Here, we provide a high-level and intuitive explanation of the proof for our main theorem, we here provide Figure 1 as a way of thinking about the proof pictorially.

The first thing we observe is that for the MOM to fail in providing a uniform error bound over the functions in $\mathcal{F}$,

there must exist a function $f \in \mathcal{F}$ for which at least half of its $\kappa$ mean estimates in the MOM fail to be $\varepsilon$-close to the true mean. However, for a fixed function $f$, we know that the MOM is likely to have almost all of its $\kappa$ mean estimates correct. We now leverage this in the first step of the analysis by introducing a "ghost" MOM, that has almost all of its $\kappa$ mean estimates correct for the function $f \in \mathcal{F}$, on which the MOM of interest had at least half of its $\kappa$ mean estimates incorrect. This step is depicted in Figure 1 as "Step 1," where the red crosses indicate whether a mean estimate is correct or not. We observe that the MOM of interest has at least half of its $\kappa$ mean estimates incorrect, whereas the "ghost" MOM has very few errors among its $\kappa$ mean estimates for the function $f \in \mathcal{F}$. This imbalance between incorrect mean estimates in the MOM of interest and the "ghost" MOM is key for "Step 3," which argues that such an imbalance is unlikely due to the symmetry introduced in this step - "Step 1" can be seen as a symmetrization of the MOM.

The next step in the analysis involves discretizing the function class $\mathcal{F}$ into a finite-sized function class $\tilde{\mathcal{F}}$. Normally, this step would be performed by creating a net over the function class $\mathcal{F}$ for any possible estimating sequence. However, since we aim to provide bounds for potentially unbounded function classes, with finite moments, we adopt an alternative discretization. Specifically, we only require the discretization $\tilde{\mathcal{F}}$ of the function class $\mathcal{F}$ to ensure that most of the mean estimates in both the MOM of interest and the "ghost" MOM remain the same - thus preserving the imbalance between incorrect mean estimates in the MOM of interest and the "ghost" MOM created in "Step 1". Furthermore, we also allow the discretization to fail for a negligible amount of mean estimates. This step is depicted as "Step 2" in Figure 1, where we observe that the discretization $\tilde{\mathcal{F}}$ of $\mathcal{F}$ preserves the imbalance between incorrect mean estimates of the MOM of interest and the "ghost" MOM.

The final step of the analysis is due to the previous two steps, to analyze the probability of the existence of a function $\tilde{f} \in \tilde{F}$ for which the MOM of interest has close to half

or more of its mean estimates incorrect, while the "ghost" MoM has very few incorrect mean estimates. First, since $\tilde{F}$ is finite, it suffices to analyze a fixed $\tilde{f} \in \tilde{F}$ and then do a union bound over $\tilde{F}$. For a fixed $\tilde{f}$, we leverage the symmetry introduced in "Step 1", namely using that the mean estimates of both the MoM of interest and the "ghost" MoM are i.i.d. Thus, we may view the $\kappa$ mean estimates of the MoM of interest and the "ghost" MoM as being "assigned" as follows: Draw two mean estimates, $\mu_{1\tilde{f},\mathbf{X}}$ and $\mu_{2\tilde{f},\mathbf{X}'}$, and with probability $1/2$, assign $\mu_{1\tilde{f},\mathbf{X}}$ to the MoM of interest and $\mu_{2\tilde{f},\mathbf{X}'}$ to the "ghost" MoM. Otherwise, assign $\mu_{2\tilde{f},\mathbf{X}'}$ to the MoM of interest and $\mu_{1\tilde{f},\mathbf{X}}$ to the "ghost" MoM. Repeat this process $\kappa$ times. Under this perspective, it is intuitively that having a large imbalance between the number of incorrect mean estimates for the MoM of interest and the "ghost" MoM - the MoM of interest has close to half or more of its mean estimates incorrect while the "ghost" MoM has very few incorrect mean estimates - is unlikely. This final step is depicted as "Step 3" in Figure 1, where the mean estimates of the MoM of interest and the "ghost" MoM are permuted.

The above high-level analysis contrasts with the conventional symmetrization-discretization-permutation argument, on the estimating sequence level, where the above analysis symmetrizes, discretizes, and permutes the mean estimates.

### 3.3. Main Result

To present our main result, we need the following definitions of discretization for a function class $\mathcal{F}$.

**Definition 3.1** (($\varepsilon, m$)-Discretization)**.** Let $0 < \varepsilon$, $m, \kappa \in \mathbb{N}$, $X_0, X_1, X_2 \in (\mathcal{X}^m)^\kappa$. A function class $\mathcal{F} \subseteq \mathbb{R}^\mathcal{X}$ admits a ($\varepsilon, m$)- **discretization** on $X_0, X_1, X_2$ if there exists a set of functions $F_{(\varepsilon, m)}$ defined on $X_0, X_1, X_2$ satisfying the following: for each $f \in \mathcal{F}$, there exists $\pi(f) \in F_{(\varepsilon, m)}$ and $I_f \subset [\kappa]$ s.t.: $|I_f| \leq \frac{2\kappa}{625}$, and for each $i \in [\kappa] \setminus I_f$ and $\forall l \in \{0, 1, 2\}$, it holds that

$$\sum_{j=1}^m \left| \frac{f(X_{l,j}^i) - \pi(f)(X_{l,j}^i)}{m} \right| \leq \varepsilon. \qquad (2)$$

We call $|F_{(\varepsilon, m)}|$ the size of the $\varepsilon$-discretization of $\mathcal{F}$ on $X_0, X_1, X_2$.

The above definition requires only that we can approximate most of the $\kappa$ sample means of a function $f \in \mathcal{F}$ appearing in its MoM with those of its neighbor $\pi(f) \in F_{(\varepsilon, m)}$, on all three samples $X_1, X_2, X_3$. The following definition extends this idea at distribution level, by requiring that with large probability, the samples $\mathbf{X}_0, \mathbf{X}_1, \mathbf{X}_2$ allows $\mathcal{F}$ to admit a ($\varepsilon, m$)-discretization.

**Definition 3.2** ($\mathcal{D}$-Discretization)**.** Let $\mathcal{D}$ be a distribution over $\mathcal{X}$. A function class $\mathcal{F} \subseteq \mathbb{R}^\mathcal{X}$ admits a $\mathcal{D}$-discretization if there exists a *threshold function* $\kappa_0 \in \mathbb{N}^{[0,1]}$, a *threshold* $\varepsilon_0 > 0$ and *size function* $N_\mathcal{D} \in \mathbb{N}^{\mathbb{R}^2}$, s.t. for any $0 < \varepsilon < \varepsilon_0$, $0 < \delta < 1$, $m \geq 1$, and $\kappa \geq \kappa_0(\delta)$, with probability at least $1 - \delta$ (over $\mathbf{X}_0, \mathbf{X}_1, \mathbf{X}_2 \sim (\mathcal{D}^m)^\kappa$) it holds that: $\mathcal{F}$ admits a ($\varepsilon, m$)-discretization $F_{(\varepsilon, m)}$ on $\mathbf{X}_0, \mathbf{X}_1, \mathbf{X}_2$ and $|F_{(\varepsilon, m)}| \leq N_\mathcal{D}(\varepsilon, m)$.

*Remark* 3.3. The following comments are in order.

- If a function class $\mathcal{F} \subseteq \mathbb{R}^\mathcal{X}$ and $\varepsilon_0 > 0$ is such that for any distribution $\mathcal{D}'$ over $\mathcal{X}$ and any $0 < \varepsilon \leq \varepsilon_0$, $\mathcal{F}$ admits a $\varepsilon$-net $\mathcal{N}_\varepsilon(\mathcal{D}', \mathcal{F}, L_1)$ in $L_1$ with respect to $\mathcal{D}'$, i.e. for any $f \in \mathcal{F}$ there exists $\pi(f) \in \mathcal{N}_\varepsilon(\mathcal{D}', \mathcal{F}, L_1)$ such that

$$\mathbb{E}_{\mathbf{X} \sim \mathcal{D}'} [|f(\mathbf{X}) - \pi(f)(\mathbf{X})|] \leq \varepsilon \qquad (3)$$

  then for any $0 < \varepsilon \leq \varepsilon_0$, $m, \kappa \in \mathbb{N}$, $X_0, X_1, X_2 \in (\mathcal{X}^m)^\kappa$, $\mathcal{F}$ admits a ($\varepsilon, m$)-discretization of size at most $\sup_{\mathcal{D}'} |\mathcal{N}_{2\varepsilon/1875}(\mathcal{D}', \mathcal{F})|$. Furthermore, for any data generating distribution $\mathcal{D}$ over $\mathbf{X}$, $\mathcal{F}$ has $\mathcal{D}$-discretization with threshold function $\kappa_0 = 1$, threshold $\varepsilon_0$ and size function $N_\mathcal{D}(\varepsilon, m) = \sup_{\mathcal{D}'} |\mathcal{N}_{2\varepsilon/1875}(\mathcal{D}', \mathcal{F})|$. See the Appendix A for a proof of this claim.

- Let $p \geq 1$. For a function class $\mathcal{F}$, it is known that the existence of a $\varepsilon$-net w.r.t. $L_p(\mathcal{D}')$ implies the existence of a $\varepsilon$-net w.r.t. $L_1(\mathcal{D}')$. Thus, each $\mathcal{F}$ admitting a net w.r.t. to the $L_p$ metric, would also admit a ($\varepsilon, m$)-discretization and a $\mathcal{D}$-discretization.

- Any function class $\mathcal{F}$ bounded between $[-1, 1]$ and featuring finite fat shattering dimension $\mathrm{FAT}_\varepsilon$ at every scale $\varepsilon > 0$, admits a $\varepsilon$-net $\mathcal{N}(\varepsilon, L_1(\mathcal{D}))$ for any $\mathcal{D}$ of size at most $\exp\left(O(\mathrm{FAT}_{O(\varepsilon)} \ln(1/\varepsilon))\right)$ (see Corollary 5.4 in Rudelson & Vershynin (2006)). This result can be also be extended to classes bounded in $[-M, M]$ for $M \geq 1$ with appropriate rescaling.

- We remark that the definition of a $\mathcal{D}$-Discretization is allowed to depend on realizations of the samples, oppose to the stricter definition of having one fixed discretization which holds for all realizations of the samples. This view of considering discretizations that depend on the samples is (to our knowledge) the most common in the literature, see e.g. (Shalev-Shwartz & Ben-David, 2014)[Definition 27.1], (Kupavskii & Zhivotovskiy, 2020)[Lemma 7] and (Rudelson & Vershynin, 2006)[Theorem 4.4 and Corollary 5.4].

We are now ready to present our main result.

**Theorem 3.4** (Main theorem)**.** *Let $\mathcal{F} \subseteq \mathbb{R}^\mathcal{X}$ and $\mathcal{D}$ be a distribution over $\mathcal{X}$. Suppose that $\mathcal{F}$ admits a $\mathcal{D}$-discretization with threshold function $\kappa_0 \in \mathbb{N}^{[0,1]}$, threshold $\varepsilon_0$, and size function $N_\mathcal{D} \in \mathbb{N}^{\mathbb{R}^2}$. Moreover, suppose that for some $p \in (1, 2]$, $\mathcal{F} \subseteq L_p(\mathcal{D})$ and let $v_p \geq$*

$\sup_{f \in \mathcal{F}} \mathbb{E}_{\mathbf{X} \sim \mathcal{D}} \left[ |f(\mathbf{X}) - \mathbb{E}_{\mathbf{X} \sim \mathcal{D}} \left[ f(\mathbf{X}) \right]|^p \right]$. *Then, there exist absolute numerical constants $c_2, c_3 > 0$ s.t., for any $\varepsilon \in (0, \varepsilon_0)$ and $\delta \in (0, 1)$, if*

$$m \geq \left( \frac{400 \cdot 16^p v_p}{\varepsilon^p} \right)^{\frac{1}{p-1}},$$

$$\kappa \geq \max \left\{ \kappa_0(\delta/8), \frac{10^6 \ln(2)}{99}, 50 \ln \left( \frac{8 N_{\mathcal{D}}(\varepsilon/16, m)}{\delta} \right) \right\},$$

*it holds*

$$\mathbb{P}_{\mathbf{X} \sim (\mathcal{D}^m)^\kappa} \left( \sup_{f \in \mathcal{F}} | \mathrm{MoM}(f, \mathbf{X}) - \mu(f)| \leq \varepsilon \right) \geq 1 - \delta. \tag{4}$$

*Remark* 3.5. The following comments are in order.

- To provide some intuition on the MoM parameters $m, \kappa, \varepsilon, \delta$, we start noting that the dependency on $\varepsilon$ decides the number of samples $m$ needed for each of the mean estimates, and is chosen such that they are within $O(\varepsilon)$ distance from the true expectation with constant probability. Furthermore, both $\varepsilon$ and $\delta$ also go into the number of mean estimates, $\kappa$. The *intuition* for the choice of $\kappa$, is that the MoM, which is based on aggregation of $\kappa$ mean estimates, boosts the constant success probability to $1 - \delta/N_{\mathcal{D}}(\varepsilon, m)$ probability for any function in the discretization, and one can then do a union bound.

- The sample complexity bound implied by our theorem is of the order of

$$\left( \frac{v_p}{\varepsilon^p} \right)^{\frac{1}{p-1}} \left( \log N_{\mathcal{D}}(\varepsilon/16, (v_p/\varepsilon^p)^{\frac{1}{p-1}}) + \log \left( \frac{1}{\delta} \right) \right),$$

when not taking into account $\kappa_0(\delta/8)$, and therefore of order $(v_p/\varepsilon^p)^{1/(p-1)} \log(v_p/(\varepsilon\delta))$ as soon as $N_{\mathcal{D}}(\varepsilon/16, (v_p/\varepsilon^p)^{1/(p-1)}) = O((v_p/\varepsilon)^\alpha)$ for some constant $\alpha$. We notice that this is optimal (up to log factors) (Devroye et al., 2016).

- In order to apply this result, one needs to find a $\mathcal{D}$-discretization of $\mathcal{F}$. In Remark 3.3 we have seen that this is possible if $\mathcal{F}$ is bounded. In the next section, we show two concrete examples of unbounded classes that admit such a cover.

- The estimation error bound in (Lecué et al., 2020), holding for $p = 2$, instead is of the order of

$$\frac{\mathcal{R}(\mathcal{F}, \mathcal{D}, n)}{n} + \sqrt{\frac{\log(1/\delta)}{n}}$$

where $n$ is the sample size and $\mathcal{R}(\mathcal{F}, \mathcal{D}, n)$ is the Rademacher complexity of $\mathcal{F}$ over a sample of size $n$.

To derive a sample complexity bound from this, one should be able to get an explicit estimate of $\mathcal{R}(\mathcal{F}, \mathcal{D}, n)$ in terms of $n$. This has already been done for certain classes of bounded or well-behaved functions (see for example (Bartlett & Mendelson, 2003; Maurer & Pontil, 2010)), it may be intersting to see if a relaxed notation of discretization, in the same spirit of Definition 3.2, can lead to explicit bounds even for broader classes of functions.

- We remark here that the magnitude of the constants in Theorem 3.4 is rather large. This is due to the symmetrization, discretization, and permutation arguments, and was not optimized. Notice that it is not uncommon for symmetrization-discretization-permutation arguments to yield large constants, for instance: (Bartlett et al., 1996) having constant of approximately 1500 (read from proof of Theorem 9 (5)), and later improved, asymptotically, by (Colomboni et al., 2025) having a constant of approximately 5000 (read from point (j) page 13), and (Long, 1999) having a constant of approximately 500 (read from lemma 9).

### 3.4. Analysis

We now give the proof of Theorem 3.4. We start by noting that for the MOM to fail, it must be the case that at least $1/2$ of the mean estimates are $\varepsilon$-away from the expectation, as in the converse case the median is $\varepsilon$-close to the expectation. Thus, to bound the failure probability of the MOM it suffices to upper bound the probability of the former event. Before presenting the upper bound, we introduce the following auxiliary random variables that will be useful throughout this section. For $\flat \in \{>, \leq\}, \kappa, m \in \mathbb{N}, \varepsilon > 0$, $\mathbf{X}_0, \mathbf{X}_1, \mathbf{X}_2 \sim ((\mathcal{D})^m)^\kappa$, $X_0, X_1, X_2 \in ((\mathcal{X})^m)^\kappa$, and a random vector $\mathbf{b} \in \{0, 1\}^\kappa$, with independent coordinates with $\mathbb{P}[\mathbf{b}_i = 0] = \mathbb{P}[\mathbf{b}_i = 1] = 1/2$, we define

$$\hat{\mathbf{S}}_{\mathbf{b}}^{(\flat)}(f, \varepsilon) = \sum_{i=1}^\kappa \frac{\mathbb{1}\{|\mu_{f, \mathbf{X}_{\mathbf{b}_i}^i} - \mu_{f, \mathbf{X}_2^i}|\flat\varepsilon\}}{\kappa},$$

$$\mathbf{S}_{\mathbf{b}}^{(\flat)}(f, \varepsilon) = \sum_{i=1}^\kappa \frac{\mathbb{1}\{|\mu_{f, X_{\mathbf{b}_i}^i} - \mu_{f, X_2^i}|\flat\varepsilon\}}{\kappa}.$$

In words $\hat{\mathbf{S}}_{\mathbf{b}}^{(>)}(f, \varepsilon)$ is the fraction of the $\kappa$ mean estimates of $f$ that are $\varepsilon$-away from the mean estimates of $f$ on the sample $\mathbf{X}_2$, and $\hat{\mathbf{S}}_{\mathbf{b}}^{(\leq)}(f, \varepsilon)$ is the fraction of the $\kappa$ mean estimates of $f$ that are $\varepsilon$-close to the mean estimates of $f$ on the sample $\mathbf{X}_2$. We also consider $\hat{\mathbf{S}}_{1-\mathbf{b}}^{(\flat)}(f, \varepsilon)$ and $\mathbf{S}_{1-\mathbf{b}}^{(\flat)}(f, \varepsilon)$, where $1 - \mathbf{b} = (1 - \mathbf{b}_1, \dots, 1 - \mathbf{b}_\kappa)$. Now we can state our symmetrization lemma.

**Lemma 3.6** (Symmetrization). *Let $p \in (1, 2], \varepsilon > 0$, and $\mathcal{D}$ a distribution over $\mathcal{X}$. Suppose that $\mathcal{F} \subseteq L_p(\mathcal{D})$, and*

*let $v_p \geq \sup_{f \in \mathcal{F}} \mathbb{E}_{\mathbf{X} \sim \mathcal{D}} \left[ |f(\mathbf{X}) - \mathbb{E}_{\mathbf{X} \sim \mathcal{D}} [f(\mathbf{X})] |^p \right]$. Then, if $m \geq \left( \frac{400 \cdot 16^p v_p}{\varepsilon^p} \right)^{\frac{1}{p-1}}$ and $\kappa \geq \left( \frac{10^6 \ln(2)}{99} \right)$ we have that*

$$
\mathbb{P}_{\mathbf{X} \sim (\mathcal{D}^m)^\kappa} \left( \exists f \in \mathcal{F} : \sum_{i=1}^{\kappa} \frac{\mathbb{1}\{|\mu_{f,\mathbf{X}^i} - \mu_f| > \varepsilon\}}{\kappa} \geq \frac{1}{2} \right)
$$

$$
\leq 4 \mathbb{P}_{\substack{\mathbf{b} \\ \mathbf{X}_0, \mathbf{X}_1, \mathbf{X}_2}} \left( \exists f \in \mathcal{F} : \hat{\mathbf{S}}_{\mathbf{b}}^{(>)} \left( f, \frac{15\varepsilon}{16} \right) \geq a, \hat{\mathbf{S}}_{1-\mathbf{b}}^{(\leq)} \left( f, \frac{2\varepsilon}{16} \right) > b \right),
$$

*where $a = \frac{4801}{10000}, b = \frac{9701}{10000}, \mathbf{b} \sim \{0,1\}^\kappa$, and $\mathbf{X}_0, \mathbf{X}_1, \mathbf{X}_2 \sim (\mathcal{D}^m)^\kappa$.*

We notice that the above lemma captures the situation described in "Step 1" of Figure 1. That is, we have related the event of the MOM failing, with the event that one MOM has many incorrect mean estimates (with the true mean being estimated), and a second MOM has few incorrect mean estimates. Notice the $\mathbf{b}_i$'s have been set up for the permutation argument, which will show that this imbalance is unlikely.

Before applying the permutation step, we discretize the function class to enable a union bound over the event that the mean estimate fails for each function in the class. The following lemma relies on the existence of a $(\varepsilon, m)$-discretization: by definition, moving from $\mathcal{F}$ to its discretization only changes the number of mean estimates that are good approximations of the "true" mean estimate $\mu_{f,\mathbf{X}_2^i}$ or, conversely, the number of bad mean estimates, slightly. In other words, this discretization preserves, the imbalance created in the symmetrization step.

**Lemma 3.7** (Discretization). *Let $m, \kappa \in \mathbb{N}, \varepsilon > 0$, and $X_0, X_1, X_2 \in (\mathcal{X}^m)^\kappa$. Suppose that $\mathcal{F}$ admits a $(\frac{\varepsilon}{16}, m)$-discretization $F_{(\varepsilon/16, m)}$ over $X_0, X_1, X_2$. Then, it holds that*

$$
\mathbb{P}_{\mathbf{b} \sim \{0,1\}^\kappa} \left( \exists f \in \mathcal{F} : \mathbf{S}_{\mathbf{b}}^{(>)} \left( f, \frac{15\varepsilon}{16} \right) \geq a, \mathbf{S}_{1-\mathbf{b}}^{(\leq)} \left( f, \frac{2\varepsilon}{16} \right) > b \right)
$$

$$
\leq |F_{(\varepsilon/16, m)}| \sup_{f \in F_{(\varepsilon/16, m)}} \mathbb{P}_{\mathbf{b} \sim \{0,1\}^\kappa} \left( \mathbf{S}_{\mathbf{b}}^{(>)} \left( f, \frac{12\varepsilon}{16} \right) \geq c,
$$

$$
\mathbf{S}_{1-\mathbf{b}}^{(>)} \left( f, \frac{12\varepsilon}{16} \right) < d \right),
$$

*where $a = \frac{4801}{10000}, b = \frac{9701}{10000}, c = \frac{4769}{10000}, d = \frac{331}{10000}$.*

The above lemma is described as "Step 2" in Figure 1. That is, the function class has been discretized while preserving the imbalance in the number of incorrect mean estimates, and the problem has now been reduced to analyzing an imbalance of correct mean estimates between two MOMs for a single function.

The following *permutation* lemma, states that having two sets of mean estimates that differ widely on their quality happens with exponentially small probability, in the number of estimates $\kappa$. This lemma models the situation depicted as "Step 3" in Figure 1.

**Lemma 3.8** (Permutation). *Let $m, \kappa \in \mathbb{N}, \varepsilon > 0$, and $X_0, X_1, X_2 \in (\mathcal{X}^m)^\kappa$. Then, for any $f \in \mathbb{R}^{\mathcal{X}}$, it holds that*

$$
\mathbb{P}_{\mathbf{b} \sim \{0,1\}^\kappa} \left( \mathbf{S}_{\mathbf{b}}^{(>)} \left( f, \frac{12\varepsilon}{16} \right) \geq c, \mathbf{S}_{1-\mathbf{b}}^{(>)} \left( f, \frac{12\varepsilon}{16} \right) < d \right)
$$

$$
\leq \exp \left( -\frac{\kappa}{50} \right)
$$

*where $c = \frac{4769}{10000}$ and $d = \frac{331}{10000}$.*

We are now ready to show the proof of Theorem 3.4.

*Proof of Theorem 3.4.* For the MOM to fail to provide a uniform estimation for $\mathcal{F}$ it must be the case that there exists a function $f \in \mathcal{F}$ s.t. at $1/2$ of the mean estimates of its MOM fails. That is

$$
\mathbb{P}_{\mathbf{X} \sim (\mathcal{D}^m)^\kappa} \left( \sup_{f \in \mathcal{F}} | \mathrm{MOM}(f, \mathbf{X}) - \mu(f)| > \varepsilon \right)
$$

$$
\leq \mathbb{P}_{\mathbf{X} \sim (\mathcal{D}^m)^\kappa} \left( \exists f \in \mathcal{F} : \sum_{i=1}^{\kappa} \frac{\mathbb{1}\{|\mu_{f,\mathbf{X}_i} - \mu_f| > \varepsilon\}}{\kappa} \geq \frac{1}{2} \right).
$$

Since $m \geq \left( \frac{400 \cdot 16^p v_p}{\varepsilon^p} \right)^{\frac{1}{p-1}}$ and $\kappa \geq \frac{10^6 \ln(2)}{99}$, Lemma 3.6 yields

$$
\mathbb{P}_{\mathbf{X} \sim (\mathcal{D}^m)^\kappa} \left( \exists f \in \mathcal{F} : | \mathrm{MOM}(f, \mathbf{X}) - \mu_f| > \varepsilon \right)
$$

$$
\leq 4 \mathbb{P}_{\substack{\mathbf{b} \sim \{0,1\}^\kappa \\ \mathbf{X}_0, \mathbf{X}_1, \mathbf{X}_2 \sim (\mathcal{D}^m)^\kappa}} \left( \exists f \in \mathcal{F} : \hat{\mathbf{S}}_{\mathbf{b}}^{(>)} \left( f, \frac{15\varepsilon}{16} \right) \geq a, \right.
$$

$$
\left. \hat{\mathbf{S}}_{1-\mathbf{b}}^{(\leq)} \left( f, \frac{2\varepsilon}{16} \right) > b \right),
$$

with $a = \frac{4801}{10000}$ and $b = \frac{9701}{10000}$. Now let $G$ denote the event that the samples $\mathbf{X}_0, \mathbf{X}_1, \mathbf{X}_2$ are s.t. $\mathcal{F}$ admits a $(\varepsilon/16, m)$-discretization of size at most $N_{\mathcal{D}}(\varepsilon/16, m)$ over them. Then, since by hypothesis $\kappa \geq \kappa_0(\delta/8)$, it holds that

$$
\mathbb{P}_{\mathbf{X} \sim (\mathcal{D}^m)^\kappa} \left( \exists f \in \mathcal{F} : | \mathrm{MOM}(f, \mathbf{X}) - \mu_f| > \varepsilon \right)
$$

$$
\leq 4 \mathbb{E}_{\mathbf{X}_0, \mathbf{X}_1, \mathbf{X}_2 \sim (\mathcal{D}^m)^\kappa} \Big[
$$

$$
\mathbb{1}\{G\} \mathbb{P}_{\mathbf{b} \sim \{0,1\}^\kappa} \left( \exists f \in \mathcal{F} : \hat{\mathbf{S}}_{\mathbf{b}}^{(>)} \left( f, \frac{15\varepsilon}{16} \right) \geq a, \right.
$$

$$
\left. \hat{\mathbf{S}}_{1-\mathbf{b}}^{(\leq)} \left( f, \frac{2\varepsilon}{16} \right) > b \right) \Big] + \delta/2. \tag{5}
$$

Since for each realization $X_0, X_1, X_2$ of $\mathbf{X}_0, \mathbf{X}_1, \mathbf{X}_2 \in G$, $\mathcal{F}$ admits a $(\varepsilon/16, m)$-discretization, Lemma 3.7 implies that

$$
\mathbb{P}_{\mathbf{b} \sim \{0,1\}^\kappa} \left( \exists f \in \mathcal{F} : \hat{\mathbf{S}}_{\mathbf{b}}^{(>)} \left( f, \frac{15\varepsilon}{16} \right) \geq a, \hat{\mathbf{S}}_{1-\mathbf{b}}^{(\leq)} \left( f, \frac{2\varepsilon}{16} \right) > b \right)
$$

$$
\leq |F_{(\varepsilon/16, m)}| \sup_{f \in F_{(\varepsilon/16, m)}} \mathbb{P}_{\mathbf{b} \sim \{0,1\}^\kappa} \left( \mathbf{S}_{\mathbf{b}}^{(>)} \left( f, \frac{12\varepsilon}{16} \right) \geq c, \right.
$$

$$
\left. \mathbf{S}_{1-\mathbf{b}}^{(>)} \left( f, \frac{12\varepsilon}{16} \right) < d \right),
$$

where $c = \frac{4769}{10000}$ and $d = \frac{331}{10000}$. Notice that the term on the right-hand side, by Lemma 3.8, can be bounded with $\exp(-\kappa/50)$. Thus, if we take $\kappa \geq 50 \ln\left(\frac{8N_{\mathcal{D}}(\varepsilon/16, m)}{\delta}\right)$, the above it is at most $\delta/8$, which, combined with (5), implies that

$$\mathbb{P}_{\mathbf{X} \sim (\mathcal{D}^m)^\kappa} (\exists f \in \mathcal{F} : |\operatorname{MoM}(f, \mathbf{X}) - \mu_f| > \varepsilon) \leq \delta$$

and concludes the proof. □

## 4. Applications

In this section we present two applications of Theorem 3.4.

### 4.1. k-Means Clustering over Unbounded Spaces

$k$-means clustering is one of the most popular clustering paradigms. Here, we provide a new sample complexity bound that improves upon existing works for the case of unbounded input and centers.

**Preliminaries.** Given $x, y \in \mathbb{R}^d$, we let $d(x, y)^2 = ||x - y||^2$. We use $k \in \mathbb{N}$ to denote the number of centers, and $Q \in \mathbb{R}^{d \times k}$ to denote the centers meant as the columns of $Q$. For $x \in \mathbb{R}^d$ and $Q \in \mathbb{R}^{d \times k}$, we let the *loss* of $Q$ on $x$ be defined as $d(x, Q)^2 = \min_{q \in Q} ||x - q||^2$, where the minimum is taken over the columns of $Q$. For a distribution $\mathcal{D}$ over $\mathbb{R}^d$, we let $\mu = \mathbb{E}_{\mathbf{X} \sim \mathcal{D}}[\mathbf{X}]$ and $\sigma^2 = \mathbb{E}_{\mathbf{X} \sim \mathcal{D}}[d(\mathbf{X}, \mu)^2]$.

In the $k$-means clustering problem, we are given random i.i.d. samples from $\mathcal{D}$, and the objective if to minimize *risk* $R(Q) = \mathbb{E}_{\mathbf{X} \sim \mathcal{D}}[d(\mathbf{X}, Q)^2]$ over $Q \in \mathbb{R}^{d \times k}$. Our goal is to provide a uniform estimation bound for all possible sets of $k$-centers. We consider the class of *normalized loss functions* defined below. For $Q \in \mathbb{R}^{d \times k}$, we define

$$f_Q(x) = \frac{2d(x, Q)^2}{\sigma^2 + \mathbb{E}_{\mathbf{X} \sim \mathcal{D}}[d(\mathbf{X}, Q)^2]},$$

and $\mathcal{F}_k = \{f_Q | Q \in \mathbb{R}^{d \times k}\}$. The class $\mathcal{F}_k$ has been introduced in Bachem et al. (2017) and provide several advantages compared to the standard loss class including scale-invariance, and that it allows to derive uniform bounds even when the support of $\mathcal{D}$ is unbounded and $Q \in \mathbb{R}^d$. The next theorem provide a bound on the sample complexity of the MoM for this problem.

**Theorem 4.1.** *Let $k \in \mathbb{N}$ and let $\mathcal{D}$ be a distribution over $\mathbb{R}^d$ s.t. $\sigma^2 < \infty$. Suppose that, there exists a $p \in (1, 2]$ s.t. $\mathcal{F}_k \subseteq L_p(\mathcal{D})$, and $\infty > v_p \geq \sup_{f \in \mathcal{F}_k} \mathbb{E}_{\mathbf{X} \sim \mathcal{D}}[|f(\mathbf{X}) - \mathbb{E}_{\mathbf{X} \sim \mathcal{D}}[f(\mathbf{X})]|^p]$. Then, $\mathcal{F}_k$ ad-*

*mits a $\mathcal{D}$-discretization with*

$$\kappa_0(\delta) = 2 \cdot 8000^2 \ln(e/\delta),$$
$$\varepsilon_0 = 1,$$
$$N_{\mathcal{D}}(\varepsilon, m) = 8\left(\frac{72 \cdot 10^4 \cdot 8000e}{\varepsilon}\right)^{140kd \ln(6k)}.$$

*Moreover, let $\varepsilon, \delta \in (0, 1)$, if*

$$m \geq \left(\frac{400 \cdot 16^p v_p}{\varepsilon^p}\right)^{\frac{1}{p-1}}$$
$$\kappa \geq \max\left(\kappa_0(\delta/8), \frac{10^6 \ln(2)}{99}, 50 \ln\left(\frac{8N_{\mathcal{D}}(\varepsilon/16, m)}{\delta}\right)\right)$$

*then*

$$\mathbb{P}_{\mathbf{x} \sim (\mathcal{D}^m)^\kappa}\left(\sup_{f \in \mathcal{F}_k} |\operatorname{MoM}(f, \mathbf{X}) - \mu(f)| \leq \varepsilon\right) \geq 1 - \delta.$$

*Remark* 4.2. The following comments are in order.

- The sample complexity bound implied by Proposition C.3 is of the order of

$$\frac{v_p^{\frac{1}{p-1}}}{\varepsilon^{\frac{p}{p-1}}}\left(dk \log k \log\frac{1}{\varepsilon} + \log\frac{1}{\delta}\right). \tag{6}$$

Notice that the $dk \log k$-term depends on the number of centers $k$ and the dimensionality of the problem $d$, and resembles a *complexity* term.

- The literature on generalization bounds for $k$-means is rich and has mostly focussed on distributions with bounded support and centers lying in a norm ball of a given radious (Linder et al., 1994; Bartlett et al., 1998; Levrard, 2013; Antos et al., 2005; Maurer & Pontil, 2010; Telgarsky & Dasgupta, 2013). The work closer to ours, in that it considers inputs and centers from unbounded sets, is Bachem et al. (2017). In that work, authors analyze the problem of uniform estimation over $\mathcal{F}_k$ with the sample mean and show a sample complexity bound of the order of

$$\frac{\mathcal{K}}{\varepsilon^2 \delta}\left(dk \log k + \log\frac{1}{\delta}\right), \tag{7}$$

where $\mathcal{K} = \mathbb{E}[d(\mathbf{X}, \mu)^4]/\sigma^4$ is the *kurtosis* of $\mathcal{D}$.

We start noticing that Bachem et al. (2017) requires the finiteness of the kurtosis, while our result only requires $\mathcal{D}$ to have a finite variance and $\mathcal{F}_k \subseteq L_p(\mathcal{D})$ for some $p \in (1, 2]$. To see that our condition is weaker, observe that when $\mathcal{K} < \infty$ then $\mathcal{F}_k \subseteq L_2(\mathcal{D})$ (See Lemma C.1 and the relation between $f \in \mathcal{F}_k$ and $s$). Focussing on the case of $p = 2$, we have the following

observations. First, notice that our sample complexity bound is exponentially better in the confidence term $1/\delta$, this is due to the stronger concentration properties of the MoM compared to the sample mean. Second, in (7) the confidence term $1/\delta$ multiplies the complexity term $dk \log k$ which is undesirable. In contrast, in our sample complexity bound these two terms are decoupled. We finally note that our bound suffers from a slightly worse dependence on $\varepsilon$ due to the extra log term.

- We have focussed on providing uniform estimation guarantees for the class $\mathcal{F}_k$ of normalized losses. In practice, one may be instead intered in bounding the risk $R(Q)$ of a certain set of centers $Q$, given its performance on the sample. Calculations show that one can get such a bound from Proposition C.3 (see the Appendix C for details). In particular, under the same assumptions of Proposition C.3, for each $Q \in \mathbb{R}^{(d \times k)}$ the following holds with probability at least $1 - \delta$

$$R(Q) \in (1 \pm \varepsilon)\Big(\text{MoM}(d(\cdot, Q)^2, \mathbf{X}) \pm \frac{\varepsilon \sigma^2}{2}\Big), \quad (8)$$

## 4.2. Linear Regression with General Losses

Linear regression is a classical problem in machine learning and statistics. This problem is typically studied either in the special case of the squared loss or for (possibly) non-smooth Lipschit losses. We consider instead the more general class of continuous losses and show a new sample complexity result that holds for broad class of distributions.

**Preliminaries.** In this section $\ell \in [0, \infty)^{\mathbb{R}}$ will denote a continuous loss function unless further specified. We consider the function class obtained composing linear predictors with bounded norm with $\ell$. That is, for $W > 0$, we define

$$\mathcal{F}_W = \left\{ \ell(\langle w, \cdot \rangle - \cdot) \mid w \in \mathbb{R}^d, ||w|| \leq W \right\}.$$

Thus, if $f \in \mathcal{F}_W$, then $f((x, y)) = \ell(\langle (w, -1), (x, y) \rangle) = \ell(\langle w, x \rangle - y)$, for any $x \in \mathbb{R}^d$, $y \in \mathbb{R}$. For $a, b > 0$, the define number $\alpha_\ell(a, b)$ as the largest positive real s.t. for $x, y \in [-a, a]$ and $|x - y| \leq \alpha_\ell(a, b)$ we have that $|\ell(x) - \ell(y)| \leq b$. Since $\ell$ is continuous and $[-a, a]$ is a compact interval, $\alpha_\ell(a, b)$ is well-defined. Thus, $\ell$ is uniform continuous on $[-a, a]$. Furthermore, when $\ell$ is $L$-Lipschitz, then $\alpha_\ell(a, b) = b/L$.

The next result provides a uniform bound that holds for general continuous losses.

**Theorem 4.3.** *Let $W > 0$ and $\mathcal{D}_X$ and $\mathcal{D}_Y$ be distributions over $\mathbb{R}^d$ and $\mathbb{R}$ respectively and let $\mathcal{D} = \mathcal{D}_X \times \mathcal{D}_Y$. Suppose that, there exists a $p \in (1, 2]$ s.t. $\mathcal{F}_W \subseteq L_p(\mathcal{D})$, and $\infty > v_p \geq \sup_{f \in \mathcal{F}_W} \mathbb{E}_{\mathbf{X} \sim \mathcal{D}} [|f(\mathbf{X}) - \mathbb{E}_{\mathbf{X} \sim \mathcal{D}} [f(\mathbf{X})]|^p].$*

*Then $\mathcal{F}_W$ admits a $\mathcal{D}$-discretization with*

$$\kappa_0(\delta) = 4 \cdot 1250^2 \ln(e/\delta),$$
$$\varepsilon_0 = \infty,$$
$$N_{\mathcal{D}}(\varepsilon, m) = \left( \frac{6W}{\beta(\varepsilon, m, \mathcal{D})} \right)^d,$$

*where*

$$\beta(\varepsilon, m, \mathcal{D}) = \min\left( \frac{W}{2}, \frac{\alpha_\ell(J, \varepsilon)}{3750 \left( \mathbb{E}[||X||_1] + \mathbb{E}[|Y|] \right) m} \right),$$
$$J = (3W/2 + 1) \cdot 3750 \left( \mathbb{E}[||\mathbf{X}||_1] + \mathbb{E}[|\mathbf{Y}|] \right) m.$$

*Moreover, let $\varepsilon \in (0, \infty)$ and $\delta \in (0, 1)$, if*

$$m \geq \left( \frac{400 \cdot 16^p v_p}{\varepsilon^p} \right)^{\frac{1}{p-1}},$$
$$\kappa \geq \max\left( \kappa_0(\delta/8), \frac{10^6 \ln(2)}{99}, 50 \ln\left( \frac{8N_{\mathcal{D}}(\varepsilon/16, m)}{\delta} \right) \right),$$

*then*

$$\mathop{\mathbb{P}}_{\mathbf{X} \sim (\mathcal{D}^m)^\kappa} \left( \sup_{f \in \mathcal{F}_W} |\text{MoM}(f, \mathbf{Z}) - \mu(f)| \leq \varepsilon \right) \geq 1 - \delta,$$

*where $\mathbf{Z} = (\mathbf{X}, \mathbf{Y}) \sim ((\mathcal{D}_X \times \mathcal{D}_Y)^m)^\kappa$.*

*Remark* 4.4. The following comments are in order.

- If we omit the dependence on $v_p$, the sample complexity bound implied by Theorem 4.3 is of the order of

$$\frac{1}{\varepsilon^{\frac{p}{p-1}}} \left( \log\left( \frac{WJ}{\alpha_\ell(J, \varepsilon)} \right) + \log\left( \frac{1}{\delta} \right) \right). \quad (9)$$

Notice that, for a given loss function $\ell$, the $d \log\left( \frac{W}{\alpha_\ell(J, \varepsilon)} \right)$ depends both on $d, \varepsilon$ and $W$ as well as $J$. Which resembles a complexity term, depending on the distribution via $J$, the complexity of $\ell$ via $\alpha_\ell$, and the norm of the regressor and its dimension $d$.

- If the loss function $\ell$ is also $L$-Lipschitz and $\ell(0) = 0$, it is possible to obtain a more explicit bound. In particular, calculations (see Appendix D for details) show that, if $\sup_{w \in \mathrm{B}(W)} \mathbb{E}[|\langle w, \mathbf{X} \rangle|^p] + \mathbb{E}[|\mathbf{Y}|^p] < \infty$, and omitting it in the following expression(also omitting $v_p$), the sample complexity is at most of the order of(assuming $W \geq 1$)

$$\frac{1}{\varepsilon^{\frac{p}{p-1}}} \left( d \log\left( \frac{WL}{\varepsilon} \right) + \log\left( \frac{1}{\delta} \right) \right). \quad (10)$$

In this case, the dependence in $\epsilon$ is explicit and of the order of $\varepsilon^{\frac{p}{1-p}} \log(1/\varepsilon)$.

- We notice that the rate 10 matches, in terms of the dependence in $\varepsilon$ and $\delta$ and up to log factors, the know rates of the sample average when the distributions of $\|\mathbf{X}\|$ and $\mathbf{Y}$ are sub-exponential (see for example Maurer & Pontil (2021)). For this class of distributions, all moments exist, while our result only requires the existence of the $p$-th moment for $p \in (1, 2]$. We also point out that a similar generality on the distribution is also achieved by Lecué et al. (2020) which also relied on the MOM estimator. The main difference is that their bound has a dependence on the Rademacher complexity of $\mathcal{F}_W$, which as far as we know, is not explicit for this class distribution.

## 5. Conclusions

In this work, we made a novel analysis of the MOM for the problem of estimating the expectation of all functions in a class only assuming that moments of order up to $p \in (1, 2]$ exist. We have focussed on the sample complexity and identified a new notation of discretization that allows the MOM to solve the task. To obtain this result, we have also developed a new symmetrization technique. We also showed that it is possible to find such a discretization in two important cases: $k$-means clustering and linear regression. It is interesting to find other applications where a discretization of the function class is possible. Finally, another problem is to match asymptotically lower and upper bounds to the sample complexity of uniform mean estimation under heavy tails, as done in Lee & Valiant (2022) for the single mean estimation problem. We leave these questions for future work.

## Acknowledgement

Mikael Møller Høgsgaard is supported by DFF Sapere Aude Research Leader Grant No. 9064-00068B by the Independent Research Fund Denmark. Andrea Paudice is supported by Novo Nordisk Fonden Start Package Grant No. NNF24OC0094365 (*Actionable Performance Guarantees in Machine Learning*).

## Impact Statement

"This paper presents work whose goal is to advance the field of Machine Learning. There are many potential societal consequences of our work, none which we feel must be specifically highlighted here."

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

# A. Details of Remark 3.3

In this appendix we elaborate on Remark 3.3.

*Remark* A.1. If a function class $\mathcal{F} \subseteq \mathbb{R}^{\mathcal{X}}$ and $\varepsilon_0 > 0$ is such that for any distribution $\mathcal{D}'$ over $\mathcal{X}$ and any $0 < \varepsilon \le \varepsilon_0$, $\mathcal{F}$ admits a $\varepsilon$-net $\mathcal{N}_\varepsilon(\mathcal{D}', \mathcal{F}, L_1)$ in $L_1$ with respect to $\mathcal{D}'$, i.e. for any $f \in \mathcal{F}$ there exists $\pi(f) \in \mathcal{N}_\varepsilon(\mathcal{D}', \mathcal{F}, L_1)$ such that

$$\mathbb{E}_{\mathbf{X} \sim \mathcal{D}'} [|f(\mathbf{X}) - \pi(f)(\mathbf{X})|] \le \varepsilon \tag{11}$$

then for any $0 < \varepsilon \le \varepsilon_0$, $m, \kappa \in \mathbb{N}$, $X_0, X_1, X_2 \in (\mathcal{X}^m)^\kappa$, $\mathcal{F}$ admits a $(\varepsilon, m)$-discretization of size at most $\sup_{\mathcal{D}'} |\mathcal{N}_{2\varepsilon/1875}(\mathcal{D}', \mathcal{F})|$. Furthermore, for any data generating distribution $\mathcal{D}$ over $\mathbf{X}$, $\mathcal{F}$ has $\mathcal{D}$-discretization with threshold function $\kappa_0 = 1$, threshold $\varepsilon_0$ and size function $N_{\mathcal{D}}(\varepsilon, m) = \sup_{\mathcal{D}'} |\mathcal{N}_{2\varepsilon/1875}(\mathcal{D}', \mathcal{F})|$

To see the above let $0 < \varepsilon \le \varepsilon_0$ $X_0, X_1, X_2 \in (\mathcal{X}^m)^\kappa$. Now consider the following distribution $\mathcal{D}'$ induced by $X_0, X_1, X_2$ given by $\mathcal{D}'(x) = \sum_{l=0}^2 \sum_{x' \in X_l} \frac{\mathbb{1}\{x = x'\}}{3\kappa m}$, i.e. points in the sequences $X_0, X_1, X_2$ are weighted proportionally to the number of occurrences they have in $X_0, X_1, X_2$. Let now $\mathcal{N}_{2\varepsilon/1875}(\mathcal{D}', \mathcal{F})$ be a $2\varepsilon/1875$-net for $\mathcal{F}$ in $L_1$ with respect to $\mathcal{D}'$ and let $I$ denote the subset of $[\kappa]$ such that $I = \{i : i \in [\kappa], \exists l \in \{0, 1, 2\}, \sum_{x \in X_l^i} \frac{|f(x) - \pi(f)(x)|}{m} > \varepsilon\}$, then we have by Equation (11) and the definition of the distribution $\mathcal{D}'$ that

$$2\varepsilon/1875 \ge \mathbb{E}_{\mathbf{X} \sim \mathcal{D}'} [|f(x) - \pi(f)(x)|] = \sum_{i=1}^k \sum_{l=0}^2 \sum_{j=1}^m \frac{|f(X_{l,j}^i) - \pi(f)(X_{l,j}^i)|}{3m\kappa}$$

$$\ge \sum_{i \in I} \sum_{l=0}^2 \sum_{j=1}^m \frac{|f(X_{l,j}^i) - \pi(f)(X_{l,j}^i)|}{3m\kappa} \ge \frac{\varepsilon|I|}{3\kappa}$$

which implies that $|I| \le \frac{6\kappa}{1875} = \frac{2\kappa}{625}$, and shows that $\mathcal{F}$ admits a $(\varepsilon, m)$-discretization $\mathcal{N}_{2\varepsilon/1875}(\mathcal{D}', \mathcal{F})$ on $X_0, X_1, X_2$, and that it has size at most $\sup_{\mathcal{D}} |\mathcal{N}_{2\varepsilon/1875}(\mathcal{D}', \mathcal{F})|$.

We notice that the above held for any $0 < \varepsilon \le \varepsilon_0$, $m, \kappa \in \mathbb{N}$, $X_0, X_1, X_2 \in (\mathcal{X}^m)^\kappa$, so especially also for the outcome of $\mathbf{X}_0, \mathbf{X}_1, \mathbf{X}_2 \sim (\mathcal{D}^m)^\kappa$ for any distribution $\mathcal{D}$ over $\mathbf{X}$ (now a data generating distribution). Thus, in the case that $\mathcal{F}$ for any distribution $\mathcal{D}'$ over $\mathcal{X}$ and any $0 < \varepsilon \le \varepsilon_0$ admits a $\varepsilon$-net $\mathcal{N}_\varepsilon(\mathcal{D}', \mathcal{F}, L_1)$ in $L_1$ with respect to $\mathcal{D}'$, then it has for any data generation distribution $\mathcal{D}$ over $\mathbf{X}$ a $\mathcal{D}$-discretization with thresholds function $\kappa_0 = 1$, (holds with probability 1 for any $\kappa \ge 1$ ), threshold $\varepsilon_0$, and size function $N_{\mathcal{D}}(\varepsilon, m) = \sup_{\mathcal{D}} |\mathcal{N}_{2\varepsilon/1875}(\mathcal{D}, \mathcal{F})|$.

We also remark that an $L_1$-net is weaker than $L_p$-nets for $p \ge 1$ (i.e. Equation (11) with $(\mathbb{E}_{\mathbf{X} \sim \mathcal{D}'} [|f(\mathbf{X}) - \pi(f)(\mathbf{X})|^p])^{1/p}$ ), so a function class $\mathcal{F}$ admitting net in $L_p$ $p \ge 1$ would also imply a $(\varepsilon, m)$-discretization of $\mathcal{F}$. Furthermore, we remark that for instance any function class which is bounded between $[-1, 1]$ and has finite fat shattering dimension $\text{FAT}_\varepsilon$ at every scale $\varepsilon > 0$ by (Rudelson & Vershynin, 2006)[Corollary 5.4] admits a $\varepsilon$-net $\mathcal{N}_\varepsilon(\mathcal{D}, \mathcal{F}, L_1)$ in $L_1$ with respect to any $\mathcal{D}$ with size $\exp\left(O(\text{FAT}_{O(\varepsilon)} \ln(1/\varepsilon))\right)$, this can also be extended to function classes bounded between $[-M, M]$ for $M \ge 1$, with suitable rescaling of the net size.

# B. Proof of lemmas used in the proof of Theorem 3.4

In this appendix we give the proof of Lemma 3.6, Lemma 3.7 and Lemma 3.8. We start with the proof of Lemma 3.6. To the end of showing Lemma 3.6 we need the following lemma which gives concentration of the sample mean given $1 < p \le 2$ central moments of the random variable $f(\mathbf{X})$. The proof of this lemma can be found Appendix E.

**Lemma B.1.** *Let $\mathcal{F} \subseteq \mathbb{R}^{\mathcal{X}}$ be a function class, $\mathcal{D}$ a distribution over $\mathcal{X}$, $1 < p \le 2$, $0 < \delta < 1$ and $0 < \varepsilon$. For $\mathcal{F} \in L_p(\mathcal{D})$,*
*$v_p = \sup_{f \in \mathcal{F}} \mathbb{E}_{\mathbf{X} \sim \mathcal{D}} [|f(\mathbf{X}) - \mathbb{E}_{\mathbf{X} \sim \mathcal{D}} [f(\mathbf{X})]|]$ and $m \ge \left(\frac{2v_p}{\delta \varepsilon^p}\right)^{\frac{1}{p-1}}$, then for any $f \in \mathcal{F}$ we have that:*

$$\mathbb{P}_{\mathbf{X} \sim \mathcal{D}^m} (|\mu_{f, \mathbf{X}} - \mu_f| > \varepsilon) \le \delta$$

With the above lemma presented, we now give the proof of Lemma 3.6

*Proof of Lemma 3.6.* To shorten the notation in the following proof we now define the following random variables,

$$\mathbf{M}_{>}(f, \varepsilon) := \sum_{i=1}^{\kappa} \frac{\mathbb{1}\{|\mu_{f, \mathbf{X}_i} - \mu_f| > \varepsilon\}}{\kappa}, \quad \check{\mathbf{M}}_{\leq}(f, \varepsilon) := \sum_{i=1}^{\kappa} \frac{\mathbb{1}\{|\mu_{f, \check{\mathbf{X}}_i} - \mu_f| \leq \varepsilon\}}{\kappa}.$$

Now notice that by the law of total probability we have

$$\mathop{\mathbb{P}}_{\mathbf{X}, \check{\mathbf{X}} \sim (\mathcal{D}^m)^{\kappa}} \left( \exists f \in \mathcal{F} : \mathbf{M}_{>}(f, \varepsilon) \geq \frac{1}{2}, \check{\mathbf{M}}_{\leq}(f, \frac{\varepsilon}{16}) > \frac{99}{100} \frac{199}{200} \right)$$

$$= \mathop{\mathbb{P}}_{\mathbf{X}, \check{\mathbf{X}} \sim (\mathcal{D}^m)^{\kappa}} \left( \exists f \in \mathcal{F} : \mathbf{M}_{>}(f, \varepsilon) \geq \frac{1}{2}, \check{\mathbf{M}}_{\leq}(f, \frac{\varepsilon}{16}) > \frac{99}{100} \frac{199}{200} \Big| \exists f \in \mathcal{F} : \mathbf{M}_{>}(f, \varepsilon) \geq \frac{1}{2} \right)$$

$$\cdot \mathop{\mathbb{P}}_{\mathbf{X}} \left( \exists f \in \mathcal{F} : \mathbf{M}_{>}(f, \varepsilon) \geq \frac{1}{2} \right) \tag{12}$$

Suppose that $\mathbf{M}_{>}(f, \varepsilon) \geq \frac{1}{2}$ for some $f \in \mathcal{F}$. Then for such $f$ and for any $i = 1, \ldots, \kappa$, we get by $m \geq \left( \frac{400 \cdot 16^p v_p}{\varepsilon^p} \right)^{\frac{1}{p-1}}$ and invoking Lemma B.1 with $\delta = 1/200$ and $\frac{\varepsilon}{16}$, that

$$\mathop{\mathbb{E}}_{\check{\mathbf{X}}_i \sim \mathcal{D}^m} \left[ \mathbb{1}\{|\mu_{f, \check{\mathbf{X}}_i} - \mu_f| > \frac{\varepsilon}{16}\} \right] = \mathbb{P}\left( |\mu_{f, \check{\mathbf{X}}_i} - \mu_f| > \frac{\varepsilon}{16} \right) \leq \frac{1}{200},$$

which implies that

$$\mathop{\mathbb{E}}_{\check{\mathbf{X}}_i \sim \mathcal{D}^m} \left[ \mathbb{1}\{|\mu_{f, \check{\mathbf{X}}_i} - \mu_f| \leq \frac{\varepsilon}{16}\} \right] > 199/200 \quad (\forall i = 1, \ldots, \kappa). \tag{13}$$

The multiplicative Chernoff inequality applied with $\kappa \geq \frac{10^6 \ln(2)}{99}$ yields

$$\mathop{\mathbb{P}}_{\check{\mathbf{X}} \sim (\mathcal{D}^m)^{\kappa}} \left( \check{\mathbf{M}}_{\leq}(f, \frac{\varepsilon}{16}) \leq \left( 1 - \frac{1}{100} \right) \mathbb{E} \check{\mathbf{M}}_{\leq}(f, \frac{\varepsilon}{16}) \right)$$

$$\leq \exp\left( -\left( \frac{1}{100} \right)^2 \kappa \mathbb{E}\left[ \mathbb{1}\{|\mu(f, \check{\mathbf{X}}) - \mu_f| \leq \frac{\varepsilon}{16}\} \right] \right)$$

$$\leq \exp\left( -\left( \frac{1}{100} \right)^2 \frac{199}{200} \kappa \right) \leq \frac{1}{2},$$

which, by (13), further implies

$$\frac{1}{2} \leq \mathop{\mathbb{P}}_{\check{\mathbf{X}} \sim (\mathcal{D}^m)^{\kappa}} \left( \check{\mathbf{M}}_{\leq}(f, \frac{\varepsilon}{16}) > \left( 1 - \frac{1}{100} \right) \mathbb{E} \check{\mathbf{M}}_{\leq}(f, \frac{\varepsilon}{16}) \right)$$

$$\leq \mathop{\mathbb{P}}_{\check{\mathbf{X}} \sim (\mathcal{D}^m)^{\kappa}} \left( \check{\mathbf{M}}_{\leq}(f, \frac{\varepsilon}{16}) > \left( 1 - \frac{1}{100} \right) \frac{199}{200} \right)$$

$$\leq \mathop{\mathbb{P}}_{\check{\mathbf{X}} \sim (\mathcal{D}^m)^{\kappa}} \left( \check{\mathbf{M}}_{\leq}(f, \frac{\varepsilon}{16}) > \frac{99}{100} \frac{199}{200} \right).$$

Thus, by independence of $\mathbf{X}$ and $\check{\mathbf{X}}$ (12) can be lower bounded by

$$\mathop{\mathbb{P}}_{\mathbf{X}, \check{\mathbf{X}} \sim (\mathcal{D}^m)^{\kappa}} \left( \exists f \in \mathcal{F} : \mathbf{M}_{>}(f, \varepsilon) \geq \frac{1}{2}, \check{\mathbf{M}}_{\leq}(f, \varepsilon) > \frac{99}{100} \frac{199}{200} \right) \tag{14}$$

$$\geq \frac{1}{2} \cdot \mathop{\mathbb{P}}_{\mathbf{X} \sim (\mathcal{D}^m)^{\kappa}} \left( \exists f \in \mathcal{F} : \mathbf{M}_{>}(f, \varepsilon) \geq \frac{1}{2} \right).$$

We now introduce a third sample $\tilde{\mathbf{X}} \sim (\mathcal{D}^m)^{\kappa}$ and for each $f \in \mathcal{F}$ and for every $\varepsilon > 0$, we define the random variables

$$\mathbf{D}_{>}(f, \varepsilon) := \sum_{i=1}^{\kappa} \frac{\mathbb{1}\{|\mu_{f, \mathbf{X}_i} - \mu_{f, \tilde{\mathbf{X}}_i}| > \varepsilon\}}{\kappa}, \quad \check{\mathbf{D}}_{\leq}(f, \varepsilon) := \sum_{i=1}^{\kappa} \frac{\mathbb{1}\{|\mu_{f, \check{\mathbf{X}}_i} - \mu_{f, \tilde{\mathbf{X}}_i}| \leq \varepsilon\}}{\kappa}.$$

We now show that

$$\mathop{\mathbb{P}}_{\mathbf{X},\check{\mathbf{X}},\tilde{\mathbf{X}}\sim(\mathcal{D}^m)^\kappa}\left(\exists f\in\mathcal{F}:\mathbf{D}_>(f,\frac{15\varepsilon}{16})>\frac{4801}{10000},\check{\mathbf{D}}_\le(f,\frac{2\varepsilon}{16})>\frac{9701}{10000}\right)$$

$$\ge\frac{1}{2}\mathop{\mathbb{P}}_{\mathbf{X},\check{\mathbf{X}}\sim(\mathcal{D}^m)^\kappa}\left(\exists f\in\mathcal{F}:\mathbf{M}_>(f,\varepsilon)\ge\frac{1}{2},\check{\mathbf{M}}_\le(f,\frac{\varepsilon}{16})>\frac{99}{100}\frac{199}{200}\right). \tag{15}$$

To this end, we again use the law of total probability to get that

$$\mathop{\mathbb{P}}_{\mathbf{X},\check{\mathbf{X}},\tilde{\mathbf{X}}\sim(\mathcal{D}^m)^\kappa}\left(\exists f\in\mathcal{F}:\mathbf{D}_>(f,\frac{15\varepsilon}{16})>\frac{4801}{10000},\check{\mathbf{D}}_\le(f,\frac{2\varepsilon}{16})>\frac{9701}{10000}\right)$$

$$=\mathop{\mathbb{P}}_{\mathbf{X},\check{\mathbf{X}},\tilde{\mathbf{X}}\sim(\mathcal{D}^m)^\kappa}\left(\exists f\in\mathcal{F}:\mathbf{D}_>(f,\frac{15\varepsilon}{16})>\frac{4801}{10000},\check{\mathbf{D}}_\le(f,\frac{2\varepsilon}{16})>\frac{9701}{10000}\right.$$

$$\left.\bigg|\exists f\in\mathcal{F}:\mathbf{M}_>(f,\varepsilon)\ge\frac{1}{2},\check{\mathbf{M}}_\le(f,\frac{\varepsilon}{16})>\left(\frac{99}{100}\right)^2\right)$$

$$\cdot\mathop{\mathbb{P}}_{\mathbf{X},\check{\mathbf{X}}\sim(\mathcal{D}^m)^\kappa}\left(\exists f\in\mathcal{F}:\mathbf{M}_>(f,\varepsilon)\ge\frac{1}{2},\check{\mathbf{M}}_\le(f,\frac{\varepsilon}{16})>\left(\frac{99}{100}\right)^2\right),$$

and show that the conditional probability term is at least $\frac{1}{2}$. Now consider a realization $X,\check{X}$ of $\mathbf{X},\check{\mathbf{X}}$ such that there exists $f\in\mathcal{F}$ where

$$\mathbf{M}_>(f,\varepsilon)\ge\frac{1}{2},\quad\check{\mathbf{M}}_\le(f,\frac{\varepsilon}{16})>\frac{99}{100}\frac{199}{200}. \tag{16}$$

Now repeating that above argument for $\check{\mathbf{X}}$ but now with $\tilde{\mathbf{X}}$ we conclude that with probability at least $1/2$ over $\tilde{\mathbf{X}}$, we have that

$$\sum_{i=1}^\kappa\frac{\mathbb{1}\{|\mu_{f,\tilde{\mathbf{X}}_i}-\mu_f|\le\frac{\varepsilon}{16}\}}{\kappa}\ge\frac{99}{100}\frac{199}{200},$$

Thus we conclude that with probability at least $1/2$ over $\tilde{\mathbf{X}}$ we have that $\mu_{f,\tilde{\mathbf{X}}_i}$ is $\frac{\varepsilon}{16}$ close to $\mu_f$ expect for a $1-\frac{99}{100}\frac{199}{200}$-fraction of $i=1,\ldots,\kappa$ so by the triangle inequality we get

$$\mathbf{M}_>(f,\varepsilon)=\sum_{i=1}^\kappa\frac{\mathbb{1}\{|\mu_{f,X_i}-\mu_f|>\varepsilon\}}{\kappa}\le\underbrace{\sum_{i=1}^\kappa\frac{\mathbb{1}\{|\mu_{f,X_i}-\mu_{f,\tilde{\mathbf{X}}_i}|>\frac{15\varepsilon}{16}\}}{\kappa}}_{\mathbf{D}_>(f,\frac{15\varepsilon}{16})}+1-\frac{99}{100}\frac{199}{200}$$

and

$$\check{\mathbf{M}}_\le(f,\varepsilon)=\sum_{i=1}^\kappa\frac{\mathbb{1}\{|\mu_{f,\check{X}_i}-\mu_f|\le\frac{\varepsilon}{16}\}}{\kappa}\le\underbrace{\sum_{i=1}^\kappa\frac{\mathbb{1}\{|\mu_{f,\check{X}_i}-\mu_{f,\tilde{\mathbf{X}}_i}|\le\frac{2\varepsilon}{16}\}}{\kappa}}_{\check{\mathbf{D}}_\le(f,\frac{2\varepsilon}{16})}+1-\frac{99}{100}\frac{199}{200}$$

which by Equation (16) implies that

$$\mathbf{D}_>(f,\frac{15}{16\varepsilon})\ge1/2-(1-\frac{99}{100}\frac{199}{200})\ge\frac{4801}{10000} \tag{17}$$

and

$$\check{\mathbf{D}}_\le(f,\frac{2}{16\varepsilon})>\frac{99}{100}\frac{199}{200}-(1-\frac{99}{100}\frac{199}{200})\ge\frac{9701}{10000} \tag{18}$$

which proves (15). Notice that, so far, we have proved that

$$\mathop{\mathbb{P}}_{\mathbf{X}\sim(\mathcal{D}^m)^\kappa}\left(\exists f\in\mathcal{F}:\mathbf{M}_>(f,\varepsilon)>\frac{1}{2}\right) \tag{19}$$

$$\leq 4\mathop{\mathbb{P}}_{\mathbf{X},\check{\mathbf{X}},\tilde{\mathbf{X}}\sim(\mathcal{D}^m)^\kappa}\left(\exists f\in\mathcal{F}:\mathbf{D}_>(f,\frac{15\varepsilon}{16})\geq\frac{4801}{10000},\check{\mathbf{D}}_\leq(f,\frac{2\varepsilon}{16})>\frac{9701}{10000}\right).$$

We now make a short digression. For $i\in[\kappa]$, let $\mathbf{X}_0^i=(\mathbf{X}_{0,1}^i,\ldots,\mathbf{X}_{0,m}^i)\sim\mathcal{D}^m$, $\mathbf{X}_1^i=(\mathbf{X}_{1,1}^i,\ldots,\mathbf{X}_{1,m}^i)\sim\mathcal{D}^m$, and $\mathbf{b}_i\sim\{0,1\}$ with $\mathbb{P}_{\mathbf{b}_i}[\mathbf{b}_i=0]=\mathbb{P}_{\mathbf{b}_i}[\mathbf{b}_i=1]=1/2$. Notice that for $A\subseteq(\mathcal{X}^m)^2$ it holds

$$\mathop{\mathbb{P}}_{\mathbf{b}_i\sim\{0,1\},\mathbf{X}_0^i\sim\mathcal{D}^m,\mathbf{X}_1^i\sim\mathcal{D}^m}\left((\mathbf{X}_{\mathbf{b}_i}^i,\mathbf{X}_{1-\mathbf{b}_i}^i)\in A\right) \tag{20}$$

$$=\frac{1}{2}\mathop{\mathbb{P}}_{\mathbf{X}_0^i\sim\mathcal{D}^m,\mathbf{X}_1^i\sim\mathcal{D}^m}\left((\mathbf{X}_1^i,\mathbf{X}_0^i)\in A\right)+\frac{1}{2}\mathop{\mathbb{P}}_{\mathbf{X}_0^i\sim\mathcal{D}^m,\mathbf{X}_1^i\sim\mathcal{D}^m}\left((\mathbf{X}_0^i,\mathbf{X}_1^i)\in A\right)$$

$$=\mathop{\mathbb{P}}_{\mathbf{X}_0^i\sim\mathcal{D}^m,\mathbf{X}_1^i\sim\mathcal{D}^m}\left((\mathbf{X}_0^i,\mathbf{X}_1^i)\in A\right) \qquad\text{(by i.i.d assumption)}$$

thus $(\mathbf{X}_{\mathbf{b}_i}^i,\mathbf{X}_{1-\mathbf{b}_i}^i)$ has the same distribution as $(\mathbf{X}_0^i,\mathbf{X}_1^i)$.

Now let $\mathbf{b}=(\mathbf{b}_1,\ldots,\mathbf{b}_\kappa)$, $\mathbf{X}=((\mathbf{X}_0^1,\mathbf{X}_1^1),\ldots,(\mathbf{X}_0^\kappa,\mathbf{X}_1^\kappa))$, $\mathbf{X}/\mathbf{X}_1:=(\mathbf{X}_2,\ldots,\mathbf{X}_k)$ and $\check{\mathbf{X}}/\check{\mathbf{X}}_1:=(\check{\mathbf{X}}_2,\ldots,\check{\mathbf{X}}_k)$. Using this notation and Equation (24), the following holds

$$\mathop{\mathbb{P}}_{\mathbf{X},\check{\mathbf{X}},\tilde{\mathbf{X}}\sim(\mathcal{D}^m)^\kappa}\left(\exists f\in\mathcal{F}:\mathbf{D}_>(f,\frac{15\varepsilon}{16})\geq\frac{4801}{10000},\check{\mathbf{D}}_\leq(f,\frac{2\varepsilon}{16})\geq\frac{9701}{10000}\right) \tag{21}$$

$$=\mathop{\mathbb{E}}_{\substack{\tilde{\mathbf{X}}\sim\mathcal{D}^m\\\mathbf{X}\backslash\mathbf{X}_1,\check{\mathbf{X}}\backslash\check{\mathbf{X}}_1\sim(\mathcal{D}^m)^{\kappa-1}}}\left[\mathop{\mathbb{P}}_{\mathbf{X}_1,\check{\mathbf{X}}_1\sim\mathcal{D}^m}\left(\exists f\in\mathcal{F}:\frac{\mathbb{1}\{|\mu_{f,\mathbf{X}_1}-\mu_{f,\tilde{\mathbf{X}}_1}|>\frac{15\varepsilon}{16}\}}{\kappa}+\sum_{i=2}^\kappa\frac{\mathbb{1}\{|\mu_{f,\mathbf{X}_i}-\mu_{f,\tilde{\mathbf{X}}_i}|>\frac{15\varepsilon}{16}\}}{\kappa}\geq\frac{4801}{10000}\right.\right.$$

$$\left.\left.,\frac{\mathbb{1}\{|\mu_{f,\check{\mathbf{X}}_1}-\mu_{f,\tilde{\mathbf{X}}_1}|\leq\frac{2\varepsilon}{16}\}}{\kappa}+\sum_{i=2}^\kappa\frac{\mathbb{1}\{|\mu_{f,\check{\mathbf{X}}_i}-\mu_{f,\tilde{\mathbf{X}}_i}|\leq\frac{2\varepsilon}{16}\}}{\kappa}>\frac{9701}{10000}\right)\right]$$

$$=\mathop{\mathbb{E}}_{\substack{\tilde{\mathbf{X}}\sim\mathcal{D}^m\\\mathbf{X}\backslash\mathbf{X}_1,\check{\mathbf{X}}\backslash\check{\mathbf{X}}_1\sim(\mathcal{D}^m)^{\kappa-1}}}\left[\mathop{\mathbb{P}}_{\substack{\mathbf{X}_0^1,\mathbf{X}_1^1\sim\mathcal{D}^m\\\mathbf{b}_1\sim\{0,1\}}}\left(\exists f\in\mathcal{F}:\frac{\mathbb{1}\{|\mu_{f,\mathbf{X}_{\mathbf{b}_1}^1}-\mu_{f,\tilde{\mathbf{X}}_1}|>\frac{15\varepsilon}{16}\}}{\kappa}+\sum_{i=2}^\kappa\frac{\mathbb{1}\{|\mu_{f,\mathbf{X}_i}-\mu_{f,\tilde{\mathbf{X}}_i}|>\frac{15\varepsilon}{16}\}}{\kappa}\geq\frac{4801}{10000}\right.\right.$$

$$\left.\left.,\frac{\mathbb{1}\{|\mu_{f,\mathbf{X}_{1-\mathbf{b}_1}^1}-\mu_{f,\tilde{\mathbf{X}}_1}|\leq\frac{2\varepsilon}{16}\}}{\kappa}+\sum_{i=2}^\kappa\frac{\mathbb{1}\{|\mu_{f,\check{\mathbf{X}}_i}-\mu_{f,\tilde{\mathbf{X}}_i}|\leq\frac{2\varepsilon}{16}\}}{\kappa}>\frac{9701}{10000}\right)\right]$$

$$\vdots \qquad\qquad\text{(repeating the above argument }\kappa-1\text{ times and renaming }\tilde{\mathbf{X}}\text{ to }\mathbf{X}_2.)$$

$$=\mathop{\mathbb{P}}_{\substack{\mathbf{b}\sim\{0,1\}^\kappa\\\mathbf{X}_0,\mathbf{X}_1,\mathbf{X}_2\sim(\mathcal{D}^m)^\kappa}}\left(\exists f\in\mathcal{F}:\sum_{i=1}^\kappa\frac{\mathbb{1}\{|\mu_{f,\mathbf{X}_{\mathbf{b}_i}^i}-\mu_{f,\mathbf{X}_2^i}|>\frac{15\varepsilon}{16}\}}{\kappa}\geq\frac{4801}{10000},\sum_{i=1}^\kappa\frac{\mathbb{1}\{|\mu_{f,\mathbf{X}_{1-\mathbf{b}_i}^i}-\mu_{f,\mathbf{X}_2^i}|\leq\frac{2\varepsilon}{16}\}}{\kappa}>\frac{9701}{10000}\right).$$

Now combining the above Equation (19) and Equation (21) we conclude that

$$\mathop{\mathbb{P}}_{\mathbf{X}\sim(\mathcal{D}^m)^\kappa}\left(\exists f\in\mathcal{F}:\mathbf{M}_>(f,\varepsilon)>\frac{1}{2}\right)$$

$$\leq 4\mathop{\mathbb{P}}_{\substack{\mathbf{b}\sim\{0,1\}^\kappa\\\mathbf{X}_0,\mathbf{X}_1,\mathbf{X}_2\sim(\mathcal{D}^m)^\kappa}}\left(\exists f\in\mathcal{F}:\sum_{i=1}^\kappa\frac{\mathbb{1}\{|\mu_{f,\mathbf{X}_{\mathbf{b}_i}^i}-\mu_{f,\mathbf{X}_2^i}|>\frac{15\varepsilon}{16}\}}{\kappa}\geq\frac{4801}{10000},\sum_{i=1}^\kappa\frac{\mathbb{1}\{|\mu_{f,\mathbf{X}_{1-\mathbf{b}_i}^i}-\mu_{f,\mathbf{X}_2^i}|\leq\frac{2\varepsilon}{16}\}}{\kappa}>\frac{9701}{10000}\right).$$

which ends the proof. $\qquad\square$

We now give the proof of Lemma 3.7

*Proof of Lemma 3.7.* We recall that $F_{\frac{\varepsilon}{16},m}$ by Equation (2), satisfies that for each $f \in \mathcal{F}$, there exists $\pi(f) \in F_{\frac{\varepsilon}{16},m}$ and $I_f \subset [\kappa]$ such that $|I_f| \leq \frac{2\kappa}{625}$ and for $i \in [\kappa] \backslash |I_f|$ it holds for each $l \in \{0, 1, 2\}$ that

$$\sum_{j=1}^{m} \left| \frac{f(X_{l,j}^i) - \pi(f)(X_{l,j}^i)}{m} \right| \leq \frac{\varepsilon}{16}. \tag{22}$$

Let for now $f \in \mathcal{F}$, and $b$ be a realization of $\mathbf{b}$. We first observe that for any $i \in [\kappa] \backslash I_f$ we have by Equation (22) that

$$|\mu_{f,X_{b_i}^i} - \mu_{f,X_2^i}| = \left| \sum_{j=1}^{m} \frac{f(X_{b_i,j}^i) - f(X_{2,j}^i)}{m} \right| \tag{23}$$

$$\leq \left| \sum_{j=1}^{m} \frac{f(X_{b_i,j}^i) - f(X_{2,j}^i) - \left( \pi(f)(X_{b_i,j}^i) - \pi(f)(X_{2,j}^i) \right)}{m} \right| + \left| \sum_{j=1}^{m} \frac{\pi(f)(X_{b_i,j}^i) - \pi(f)(X_{2,j}^i)}{m} \right|$$

$$\text{(by triangle inequality)}$$

$$\leq \sum_{j=1}^{m} \frac{|f(X_{b_i,j}^i) - \pi(f)(X_{b_i,j}^i)|}{m} + \sum_{j=1}^{m} \frac{|f(X_{2,j}^i) - \pi(f)(X_{2,j}^i)|}{m} + |\mu_{\pi(f),X_{b_i}^i} - \mu_{\pi(f),X_2^i}| \quad \text{(by triangle inequality)}$$

$$\leq \frac{2\varepsilon}{16} + |\mu_{\pi(f),X_{b_i}^i} - \mu_{\pi(f),X_2^i}|. \qquad \text{(by } i \in [\kappa] \backslash I_f \text{ and Equation (22))}$$

Similarly we conclude that for any $i \in [\kappa] \backslash I_f$ we have that

$$|\mu_{f,X_{1-b_i}^i} - \mu_{f,X_2^i}| = \left| \sum_{j=1}^{m} \frac{f(X_{1-b_i,j}^i) - f(X_{2,j}^i)}{m} \right| \tag{24}$$

$$\geq \left| \sum_{j=1}^{m} \frac{\pi(f)(X_{1-b_i,j}^i) - \pi(f)(X_{2,j}^i)}{m} \right| - \left| \sum_{j=1}^{m} \frac{f(X_{1-b_i,j}^i) - f(X_{2,j}^i) - \left( \pi(f)(X_{1-b_i,j}^i) - \pi(f)(X_{2,j}^i) \right)}{m} \right|$$

$$\text{(by reverse triangle inequality)}$$

$$\geq |\mu_{\pi(f),X_{1-b_i}^i} - \mu_{\pi(f),X_2^i}| - \sum_{j=1}^{m} \frac{|f(X_{1-b_i,j}^i) - \pi(f)(X_{1-b_i,j}^i)|}{m} - \sum_{j=1}^{m} \frac{|f(X_{2,j}^i) - \pi(f)(X_{2,j}^i)|}{m}$$

$$\text{(by triangle inequality)}$$

$$\geq |\mu_{\pi(f),X_{1-b_i}^i} - \mu_{\pi(f),X_2^i}| - \frac{2\varepsilon}{16}. \qquad \text{(by } i \in [\kappa] \backslash I_f \text{ and Equation (22))}$$

Now using Equation (23) we get that

$$\sum_{i=1}^{\kappa} \frac{\mathbb{1}\{|\mu_{f,X_{b_i}^i} - \mu_{f,X_2^i}| > \frac{15\varepsilon}{16}\}}{\kappa} \tag{25}$$

$$= \sum_{i \notin I_f} \frac{\mathbb{1}\{|\mu_{f,X_{b_i}^i} - \mu_{f,X_2^i}| > \frac{15\varepsilon}{16}\}}{\kappa} + \sum_{j \in I_f} \frac{\mathbb{1}\{|\mu_{f,X_{b_i}^i} - \mu_{f,X_2^i}| > \frac{15\varepsilon}{16}\}}{\kappa}$$

$$\leq \sum_{i \notin I_f} \frac{\mathbb{1}\{|\mu_{f,X_{b_i}^i} - \mu_{f,X_2^i}| > \frac{15\varepsilon}{16}\}}{\kappa} + \sum_{i \in I_f} \frac{\mathbb{1}\{|\mu_{\pi(f),X_{b_i}^i} - \mu_{\pi(f),X_2}| > \frac{13\varepsilon}{16}\}}{\kappa} + \frac{2}{625} \qquad \text{(by } |I_f| \leq \frac{2\kappa}{625})$$

$$\leq \sum_{i \notin I_f} \frac{\mathbb{1}\{|\mu_{\pi(f),X_{b_i}^i} - \mu_{\pi(f),X_2}| > \frac{13\varepsilon}{16}\}}{\kappa} + \sum_{i \in I_f} \frac{\mathbb{1}\{|\mu_{\pi(f),X_{b_i}^i} - \mu_{\pi(f),X_2}| > \frac{13\varepsilon}{16}\}}{\kappa} + \frac{2}{625}. \qquad \text{(by Equation (23))}$$

$$= \sum_{i=1}^{\kappa} \frac{\mathbb{1}\{|\mu_{\pi(f),X_{b_i}^i} - \mu_{\pi(f),X_2}| > \frac{13\varepsilon}{16}\}}{\kappa} + \frac{2}{625}.$$

Furthermore using Equation (24) we get that

$$\sum_{i=1}^{\kappa} \frac{\mathbb{1}\{|\mu_{f,X_{1-b_i}^i} - \mu_{f,X_2^i}| \le \frac{2\varepsilon}{16}\}}{\kappa} \tag{26}$$

$$= \sum_{i \notin I_f} \frac{\mathbb{1}\{|\mu_{f,X_{1-b_i}^i} - \mu_{f,X_2^i}| \le \frac{2\varepsilon}{16}\}}{\kappa} + \sum_{i \in I_f} \frac{\mathbb{1}\{|\mu_{f,X_{1-b_i}^i} - \mu_{f,X_2^i}| \le \frac{2\varepsilon}{16}\}}{\kappa}$$

$$\le \sum_{i \notin I_f} \frac{\mathbb{1}\{|\mu_{f,X_{1-b_i}^i} - \mu_{f,X_2^i}| \le \frac{2\varepsilon}{16}\}}{\kappa} + \sum_{i \in I_f} \frac{\mathbb{1}\{|\mu_{\pi(f),X_{1-b_i}^i} - \mu_{\pi(f),X_2}| \le \frac{4\varepsilon}{16}\}}{\kappa} + \frac{2}{625} \qquad \text{(by } |I_f| \le \frac{2\kappa}{625} \text{)}$$

$$\le \sum_{i \notin I_f} \frac{\mathbb{1}\{|\mu_{\pi(f),X_{1-b_i}^i} - \mu_{\pi(f),X_2}| \le \frac{4\varepsilon}{16}\}}{\kappa} + \sum_{i \in I_f} \frac{\mathbb{1}\{|\mu_{\pi(f),X_{1-b_i}^i} - \mu_{\pi(f),X_2}| \le \frac{4\varepsilon}{16}\}}{\kappa} + \frac{2}{625} \qquad \text{(by Equation (24))}$$

$$= \sum_{i=1}^{\kappa} \frac{\mathbb{1}\{|\mu_{\pi(f),X_{1-b_i}^i} - \mu_{\pi(f),X_2}| \le \frac{4\varepsilon}{16}\}}{\kappa} + \frac{2}{625}$$

Since we showed the above for any $f \in \mathcal{F}$ we notice that this implies that

$$\exists f \in \mathcal{F} : \sum_{i=1}^{\kappa} \frac{\mathbb{1}\{|\mu_{f,X_{b_i}^i} - \mu_{f,X_2^i}| > \frac{15\varepsilon}{16}\}}{\kappa} \ge \frac{4801}{10000}, \sum_{i=1}^{\kappa} \frac{\mathbb{1}\{|\mu_{f,X_{1-b_i}^i} - \mu_{f,X_2^i}| \le \frac{2\varepsilon}{16}\}}{\kappa} > \frac{9701}{10000}$$

$$\Rightarrow \exists f \in \mathcal{F} : \sum_{i=1}^{\kappa} \frac{\mathbb{1}\{|\mu_{\pi(f),X_{b_i}^i} - \mu_{\pi(f),X_2}| > \frac{13\varepsilon}{16}\}}{\kappa} \ge \frac{4769}{10000}, \sum_{i=1}^{\kappa} \frac{\mathbb{1}\{|\mu_{\pi(f),X_{1-b_i}^i} - \mu_{\pi(f),X_2}| \le \frac{4\varepsilon}{16}\}}{\kappa} > \frac{9669}{10000}$$

$$\Rightarrow \exists f \in F_{\frac{\varepsilon}{16},m} : \sum_{i=1}^{\kappa} \frac{\mathbb{1}\{|\mu_{f,X_{b_i}^i} - \mu_{f,X_2^i}| > \frac{12\varepsilon}{16}\}}{\kappa} \ge \frac{4769}{10000}, \sum_{i=1}^{\kappa} \frac{\mathbb{1}\{|\mu_{f,X_{1-b_i}^i} - \mu_{f,X_2^i}| > \frac{12\varepsilon}{16}\}}{\kappa} \le \frac{331}{10000}.$$

Since we showed the above for any realization $b$ of $\mathbf{b}$ it also holds for random $\mathbf{b}$, and thus we conclude by the union bound that

$$\mathop{\mathbb{P}}_{\mathbf{b} \sim \{0,1\}^\kappa} \Big( \exists f \in \mathcal{F} : \sum_{i=1}^{\kappa} \frac{\mathbb{1}\{|\mu_{f,X_{\mathbf{b}_i}^i} - \mu_{f,X_2^i}| > \frac{15\varepsilon}{16}\}}{\kappa} \ge \frac{4801}{10000}, \sum_{i=1}^{\kappa} \frac{\mathbb{1}\{|\mu_{f,X_{1-\mathbf{b}_i}^i} - \mu_{f,X_2^i}| \le \frac{2\varepsilon}{16}\}}{\kappa} > \frac{9701}{10000} \Big)$$

$$\le \mathop{\mathbb{P}}_{\mathbf{b} \sim \{0,1\}^\kappa} \Big( \exists f \in F_{\frac{\varepsilon}{16},m} : \sum_{i=1}^{\kappa} \frac{\mathbb{1}\{|\mu_{f,X_{\mathbf{b}_i}^i} - \mu_{f,X_2^i}| > \frac{12\varepsilon}{16}\}}{\kappa} \ge \frac{4769}{10000}, \sum_{i=1}^{\kappa} \frac{\mathbb{1}\{|\mu_{f,X_{1-\mathbf{b}_i}^i} - \mu_{f,X_2^i}| > \frac{12\varepsilon}{16}\}}{\kappa} \le \frac{331}{10000} \Big)$$

$$\le \sum_{f \in F_{\frac{\varepsilon}{16},m}} \mathop{\mathbb{P}}_{\mathbf{b} \sim \{0,1\}^\kappa} \Big( \sum_{i=1}^{\kappa} \frac{\mathbb{1}\{|\mu_{f,X_{\mathbf{b}_i}^i} - \mu_{f,X_2^i}| > \frac{12\varepsilon}{16}\}}{\kappa} \ge \frac{4769}{10000}, \sum_{i=1}^{\kappa} \frac{\mathbb{1}\{|\mu_{f,X_{1-\mathbf{b}_i}^i} - \mu_{f,X_2^i}| > \frac{12\varepsilon}{16}\}}{\kappa} \le \frac{331}{10000} \Big)$$

$$\le |F_{\frac{\varepsilon}{16},m}| \sup_{f \in F_{\frac{\varepsilon}{16},m}} \mathop{\mathbb{P}}_{\mathbf{b} \sim \{0,1\}^\kappa} \Big( \sum_{i=1}^{\kappa} \frac{\mathbb{1}\{|\mu_{f,X_{\mathbf{b}_i}^i} - \mu_{f,X_2^i}| > \frac{12\varepsilon}{16}\}}{\kappa} \ge \frac{4769}{10000}, \sum_{i=1}^{\kappa} \frac{\mathbb{1}\{|\mu_{f,X_{1-\mathbf{b}_i}^i} - \mu_{f,X_2^i}| > \frac{12\varepsilon}{16}\}}{\kappa} \le \frac{331}{10000} \Big),$$

which finishes the proof. $\qquad \square$

We now give the proof of Lemma 3.8.

*Proof of Lemma 3.8.* Let $\zeta$ be the following fixed number

$$\zeta = \sum_{i=1}^{\kappa} \mathbb{1}\{|\mu_{f,X_{\mathbf{b}_i}^i} - \mu_{f,X_2^i}| > \frac{12\varepsilon}{16}\} + \sum_{i=1}^{\kappa} \mathbb{1}\{|\mu_{f,X_{1-\mathbf{b}_i}^i} - \mu_{f,X_2^i}| > \frac{12\varepsilon}{16}\}$$

$$= \sum_{j=0}^{1} \sum_{i=1}^{\kappa} \mathbb{1}\{|\mu_{f,X_j^i} - \mu_{f,X_2^i}| > \frac{12\varepsilon}{16}\}.$$

We notice that for the probability in Lemma 3.8 to be greater than $0$, it must be the case that $\zeta \geq \frac{4769k}{10000}$, from now on assume this is the case. Furthermore, we notice that since $\zeta \geq \frac{4769k}{10000}$, there is at least $\frac{4769k}{20000}$, $i$'s such that either $\mu_{f,X_0^i}$ or $\mu_{f,X_1^i}$ is $\frac{12\varepsilon}{16}$ away from $\mu_{f,X_2^i}$, i.e. $|\mu_{f,X_0^i} - \mu_{f,X_2^i}| > \frac{12\varepsilon}{16}$ or $|\mu_{f,X_1^i} - \mu_{f,X_2^i}| > \frac{12\varepsilon}{16}$, let $I \subset \{1,\ldots,\kappa\}$ be the set of these indexes, i.e. $I = \{i \in [\kappa] : |\mu_{f,X_0^i} - \mu_{f,X_2^i}| > \frac{12\varepsilon}{16} \text{ or } |\mu_{f,X_1^i} - \mu_{f,X_2^i}| > \frac{12\varepsilon}{16}\}$. We notice that for $i \in I$ we have with probability at least $\frac{1}{2}$ that $|\mu_{1-\mathbf{b}_i}(f) - \mu_{f,X_2^i}| > \frac{12\varepsilon}{16}$, furthermore we have that the random variable $\sum_{i=1}^\kappa \frac{\mathbb{1}\{|\mu_{1-\mathbf{b}_i}(f) - \mu_{f,X_2^i}| > \frac{12\varepsilon}{16}\}}{\kappa} = \sum_{i \in I} \frac{\mathbb{1}\{|\mu_{1-\mathbf{b}_i}(f) - \mu_{f,X_2^i}| > \frac{12\varepsilon}{16}\}}{\kappa}$, is a sum of independent $\{0,1\}$-random variables. We notice that by $|I| \geq \frac{4769k}{20000}$ this random variable has expectation

$$\eta := \mathop{\mathbb{E}}_{\mathbf{b} \sim \{0,1\}^\kappa} \left[ \sum_{i \in I} \frac{\mathbb{1}\{|\mu_{1-\mathbf{b}_i}(f) - \mu_{f,X_2^i}| > \frac{12\varepsilon}{16}\}}{\kappa} \right] \geq \sum_{i \in I} \frac{1}{2k} \geq \frac{4769}{40000}.$$

We notice that

$$\frac{331}{10000} = (1 - \frac{\eta - \frac{331}{10000}}{\eta})\eta,$$

and since $\frac{x - \frac{331}{10000}}{x}$ is an increasing function for $x \geq \frac{331}{10000}$ (has derivative $\frac{\frac{331}{10000}}{x^2} > 0$), and $\eta \geq \frac{4769}{40000} > \frac{331}{10000}$, we conclude that $1 > \frac{\eta - \frac{331}{10000}}{\eta} \geq \frac{\frac{4769}{40000} - \frac{331}{10000}}{\frac{4769}{40000}} = \frac{3445}{4769}$. Thus, by an application of a multiplicative Chernoff bound we get that

$$\mathop{\mathbb{P}}_{\mathbf{b} \sim \{0,1\}^\kappa} \left[ \sum_{i=1}^\kappa \frac{\mathbb{1}\{|\mu_{1-\mathbf{b}_i}(f) - \mu_{f,X_2^i}| > \frac{12\varepsilon}{16}\}}{\kappa} \leq \frac{331}{10000} \right]$$

$$= \mathop{\mathbb{P}}_{\mathbf{b} \sim \{0,1\}^\kappa} \left[ \sum_{i \in I} \frac{\mathbb{1}\{|\mu_{1-\mathbf{b}_i}(f) - \mu_{f,X_2^i}| > \frac{12\varepsilon}{16}\}}{\kappa} \leq (1 - \frac{\eta - \frac{331}{10000}}{\eta})\eta \right]$$

$$\leq \exp\left( -\frac{\left(\frac{\eta - \frac{331}{10000}}{\eta}\right)^2 \eta\kappa}{2} \right) \leq \exp\left( -\frac{\left(\frac{3445}{4769}\right)^2 \eta\kappa}{2} \right) \qquad \text{(by } \frac{\eta - \frac{331}{10000}}{\eta} \geq \frac{3445}{4769}\text{)}$$

$$\leq \exp\left( -\frac{474721}{15260800}\kappa \right) \leq \exp(-\kappa/50), \qquad \text{(by } \eta \geq \frac{4769}{40000}\text{,)}$$

where the last inequality follows from $\frac{474721}{15260800} \geq \frac{1}{50}$, and concludes the proof. $\qquad \square$

## C. k -means

In this appendix we give the proof of the sample complexity result for the $k$-mean's objective. To this end, we first introduce some preliminaries and some results from (Bachem et al., 2017), which we will use in the following proof.

### C.1. Preliminaries

We follow the notation used in (Bachem et al., 2017). For two points $x, y \in \mathbb{R}^d$ we write $d(x,y)^2 = ||x - y||^2$, for a point $x \in \mathbb{R}^d$ and $Q \subset \mathbb{R}^d$ we let $d(x,Q)^2 = \min_{q \in Q} ||x - q||^2$. We use $k \in \mathbb{N}$, for being the number of centers written $Q \in \mathbb{R}^{d \times k}$, whereby we mean that the columns of $Q$ are the centers. For a distribution $\mathcal{D}$ over $\mathbb{R}^d$ we define $\mu = \mathbb{E}_{\mathbf{X} \sim \mathcal{D}}[\mathbf{X}]$ and $\sigma^2 = \mathbb{E}_{\mathbf{X} \sim \mathcal{D}}\left[d(\mathbf{X}, \mu)^2\right]$. For a $Q \in \mathbb{R}^{d \times k}$ we define the following function $f_Q$ given by $\frac{2d(x,Q)^2}{\sigma^2 + \mathbb{E}_{\mathbf{X} \sim \mathcal{D}}[d(\mathbf{X},Q)^2]}$. For $k \in \mathbb{N}$ we will use $\mathcal{F}_k$ to denote the function class $\mathcal{F}_k = \{f_Q | Q \in \mathbb{R}^{d \times k}\}$.

To describe the complexity of $\mathcal{F}_k$ in the following we will need the definition of the pseudo dimension of a function class $\mathcal{F} \subseteq \mathbb{R}^{\mathcal{X}}$. The pseudo dimension is defined as the largest number $Pdim(\mathcal{F}) = d$, such that there exists a point set $x_1, \ldots, x_d$ and thresholds $r_1, \ldots, r_d$, where for any $b \in \{0,1\}^d$, there exist a function $f \in \mathcal{F}$ such that for $i \in [d]$ and $b_i = 0$, then $f$ is below $r_i$ i.e. $f(x_i) < r_i$ and if $b_i = 1$ then $f$ is above $r_i$ i.e. $f(x_i) \geq r_i$. We now introduce two lemmas from (Bachem et al., 2017) that we are going to need in the following the first lemma states some useful properties about $f_Q$

**Lemma C.1** (Lemma 1 in (Bachem et al., 2017) ). *Let* $k \in \mathbb{N}$, $\mathcal{D}$ *a distribution on* $\mathbb{R}^d$, *with* $\mu = \mathbb{E}_{\mathbf{X} \sim \mathcal{D}}[\mathbf{X}]$ *and* $\sigma^2 = \mathbb{E}_{\mathbf{X} \sim \mathcal{D}}\left[d(\mathbf{X}, \mu)^2\right]$. *For any* $Q \in \mathbb{R}^{d \times k}$ *define* $f_Q(x) = \frac{2d(x,Q)^2}{\sigma^2 + \mathbb{E}_{\mathbf{X} \sim \mathcal{D}}[d(\mathbf{X},Q)^2]}$, *the function class* $\mathcal{F}_k = \{f_Q | Q \in \mathbb{R}^{d \times k}\}$

and $s(x) = \frac{4d(x,\mu)^2}{\sigma^2} + 8$. We then have that $Pdim(\mathcal{F}) \leq 6k(d+4)\ln(6k)/\ln(2)$ and for all $x \in \mathbb{R}^d$ and $Q \in \mathbb{R}^{d \times k}$ we have that $f_Q(x) \leq s(x)$.

The next lemma roughly says that when $s(x)$ is bounded on a point set, then one can bound the size of a maximal $\varepsilon$-packing of $\mathcal{F}_k$. As in (Bachem et al., 2017) we state the result for a general function class $\mathcal{F}$. For this we need to define what we mean by a maximal packing, to this end let $\mathcal{D}$ be a distribution on $\mathcal{X}$ and $\mathcal{F} \subseteq [0,\infty]^{\mathcal{X}}$, we then say that $P_\varepsilon \subseteq \mathcal{F}$ is a $\varepsilon$-packing of $\mathcal{F}$ in $L_1(\mathcal{D})$ if for any $f, g \in P_\varepsilon$ where $f \neq g$ we have that

$$\mathbb{E}_{\mathbf{X} \sim \mathcal{D}}[|f(\mathbf{X}) - g(\mathbf{X})|] > \varepsilon. \tag{27}$$

The packing $P_\varepsilon$ is maximal for $\mathcal{F}$ if it has the largest size possible.

**Lemma C.2** (Lemma 4 in (Bachem et al., 2017)). *Let $\mathcal{F} \subseteq [0,\infty]^{\mathcal{X}}$ be a function class with $d = Pdim(\mathcal{F}) < \infty$ and for $x \in \mathcal{X}$ define $s(x) = \sup_{x \in \mathcal{X}} f(x)$. Let $\mathcal{D}$ be a distribution on $\mathcal{X}$, such that $0 < \mathbb{E}_{\mathbf{X} \sim \mathcal{D}}[s(\mathbf{X})] < \infty$. It then holds for any $0 < \varepsilon \leq \mathbb{E}_{\mathbf{X} \sim \mathcal{D}}[s(\mathbf{X})]$ that any maximal $\varepsilon$-packing $P_\varepsilon$ of $\mathcal{F}$ in $L_1(\mathcal{D})$ has size at most $8(2e\,\mathbb{E}_{\mathbf{X} \sim \mathcal{D}}[s(\mathbf{X})]/\varepsilon)^{2d}$.*

## C.2. Proof of Proposition C.3

We now state our Proposition C.3 about the k-means objective introduced above.

**Proposition C.3** ( k -means). *Let $k \in \mathbb{N}$, $0 < \delta, \varepsilon < 1$, $\mathcal{D}$ be a distribution over $\mathbb{R}^d$, $\mu = \mathbb{E}_{\mathbf{X} \sim \mathcal{D}}[\mathbf{X}]$, $1 < p \leq 2$ and $\mathcal{F}_k = \{f_Q \mid Q \in \mathbb{R}^{d \times k}\}$ if $\sigma^2 = \mathbb{E}_{\mathbf{X} \sim \mathcal{D}}[d(\mathbf{X}, \mu)^2] < \infty$, $\mathcal{F}_k \in L_p(\mathcal{D})$ and $v_p \geq \sup_{f \in \mathcal{F}_k} \mathbb{E}_{\mathbf{X} \sim \mathcal{D}}[|f(\mathbf{X}) - \mathbb{E}_{\mathbf{X} \sim \mathcal{D}}[f(\mathbf{X})]|^p]$ then for $m \geq \left(\frac{400 \cdot 16^p v_p}{\varepsilon^p}\right)^{\frac{1}{p-1}}$ and $\kappa \geq \max\left(\kappa_0(\delta/8), \frac{10^6 \ln(2)}{99}, 50\ln\left(\frac{8N(\mathcal{D}, \varepsilon/16, m)}{\delta}\right)\right)$ where $\kappa_0(\delta) = 2 \cdot 8000^2 \ln(e/\delta)$ and $N(\mathcal{D}, \varepsilon, m) = 8\left(\frac{72 \cdot 10^4 \cdot 8000e}{\varepsilon}\right)^{140kd \ln(6k)}$ we have that with probability at least $1 - \delta$ over $\mathbf{X} \sim (\mathcal{D}^m)^\kappa$: for all $f \in \mathcal{F}_k$ that*

$$|\text{MoM}(f, \mathbf{X}) - \mathbb{E}_{\mathbf{X}' \sim \mathcal{D}}[f(\mathbf{X}')]| \leq \varepsilon$$

Before we give the proof of Proposition C.3 we make the following observation. Proposition C.3 considers the functions $f_Q(x) = \frac{2d(x,Q)^2}{\sigma^2 + \mathbb{E}_{\mathbf{X} \sim \mathcal{D}}[d(\mathbf{X}, Q)^2]}$. For a fixed $Q$, the denominator in $f_Q$ is a fixed positive number, whereby we get that the median of the means $\mu_{f_Q, \mathbf{X}^i} = \sum_{j=1}^m \frac{2d(\mathbf{X}_j^i, Q)^2}{(\sigma^2 + \mathbb{E}_{\mathbf{X} \sim \mathcal{D}}[d(\mathbf{X}, Q)^2])m}$ is just the same as scaling the median of the means $\sum_{j=1}^m \frac{d(\mathbf{X}_j^i, Q)^2}{m}$ with $2/\left(\sigma^2 + \mathbb{E}_{\mathbf{X} \sim \mathcal{D}}[d(\mathbf{X}, Q)^2]\right)$. Whereby we conclude that

$$|\text{MoM}(f, \mathbf{X}) - \mathbb{E}_{\mathbf{X}' \sim \mathcal{D}}[f(\mathbf{X}')]| \leq \varepsilon$$

implies by multiplication of $2/\left(\sigma^2 + \mathbb{E}_{\mathbf{X} \sim \mathcal{D}}[d(\mathbf{X}, Q)^2]\right)$ that

$$|\text{MoM}(d(\cdot, Q)^2, \mathbf{X}) - \mathbb{E}_{\mathbf{X}' \sim \mathcal{D}}[d(\mathbf{X}', Q)^2]| \leq \frac{\varepsilon\left(\sigma^2 + \mathbb{E}_{\mathbf{X}' \sim \mathcal{D}}[d(\mathbf{X}', Q)^2]\right)}{2}, \tag{28}$$

and furthermore by rearrangement that

$$\mathbb{E}_{\mathbf{X}' \sim \mathcal{D}}[d(\mathbf{X}', Q)^2] \leq \left(\frac{1}{1 - \varepsilon/2}\right)\left(\text{MoM}(d(\cdot, Q,)\mathbf{X})^2) + \frac{\varepsilon\sigma^2}{2}\right) \leq (1 + \varepsilon)\left(\text{MoM}(d(\cdot, Q,)\mathbf{X})^2) + \frac{\varepsilon\sigma^2}{2}\right),$$

where the last inequality follows from $1 + \frac{x/2}{1 - x/2} \leq 1 + x$ for $0 \leq x \leq 1$, and again by rearrangement of Equation (28) we get that

$$\mathbb{E}_{\mathbf{X}' \sim \mathcal{D}}[d(\mathbf{X}', Q)^2] \geq \left(\frac{1}{1 + \varepsilon/2}\right)\left(\text{MoM}(d(\cdot, Q)^2, \mathbf{X}) - \frac{\varepsilon\sigma^2}{2}\right)$$

which implies if the term $\left(\text{MoM}(d(\cdot, Q)^2, \mathbf{X}) - \frac{\varepsilon\sigma^2}{2}\right) \geq 0$ that

$$\mathbb{E}_{\mathbf{X}' \sim \mathcal{D}}[d(\mathbf{X}', Q)^2] \geq (1 - \varepsilon)\left(\text{MoM}(d(\cdot, Q)^2, \mathbf{X}) - \frac{\varepsilon\sigma^2}{2}\right),$$

by $\frac{1}{1 + \varepsilon/2} \geq 1 - \varepsilon$, but also holds in the case that the term $\left(\text{MoM}(d(\cdot, Q)^2, \mathbf{X}) - \frac{\varepsilon\sigma^2}{2}\right) < 0$ since $\mathbb{E}_{\mathbf{X}' \sim \mathcal{D}}[d(\mathbf{X}', Q)^2]$ is non-negative and $\varepsilon \leq 1$. We compile these observations in the following corollary

**Corollary C.4** (k-means). *Let $k \in \mathbb{N}$, $0 < \delta, \varepsilon < 1$, $\mathcal{D}$ be a distribution over $\mathbb{R}^d$, $\mu = \mathbb{E}_{\mathbf{X} \sim \mathcal{D}}[\mathbf{X}]$, $1 < p \leq 2$ and $\mathcal{F}_k = \left\{ f_Q \mid Q \in \mathbb{R}^{d \times k} \right\}$ if $\sigma^2 = \mathbb{E}_{\mathbf{X} \sim \mathcal{D}}[d(\mathbf{X}, \mu)^2] < \infty$, $\mathcal{F}_k \in L_p(\mathcal{D})$ and $v_p \geq \sup_{f \in \mathcal{F}_k} \mathbb{E}_{\mathbf{X} \sim \mathcal{D}}[|f(\mathbf{X}) - \mathbb{E}_{\mathbf{X} \sim \mathcal{D}}[f(\mathbf{X})]|^p]$ then for $m \geq \left( \frac{400 \cdot 16^p v_p}{\varepsilon^p} \right)^{\frac{1}{p-1}}$ and $\kappa \geq \max \left( \kappa_0(\delta/8), \frac{10^6 \ln(2)}{99}, 50 \ln \left( \frac{8N(\mathcal{D}, \varepsilon/16, m)}{\delta} \right) \right)$ where $\kappa_0(\delta) = 2 \cdot 8000^2 \ln(e/\delta)$ and $N(\mathcal{D}, \varepsilon, m) = 8 \left( \frac{72 \cdot 10^4 \cdot 8000 e}{\varepsilon} \right)^{140kd \ln(6k)}$ we then have with probability at least $1 - \delta$ over $\mathbf{X} \sim (\mathcal{D}^m)^\kappa$ that: for all $Q \subset \mathbb{R}^d$ such that $|Q| = k$*

$$(1 - \varepsilon) \left( \text{MoM}(d(\cdot, Q,)\mathbf{X})^2) - \frac{\varepsilon \sigma^2}{2} \right) \leq \mathbb{E}_{\mathbf{X}' \sim \mathcal{D}} \left[ d(\mathbf{X}', Q)^2 \right] \leq (1 + \varepsilon) \left( \text{MoM}(d(\cdot, Q,)\mathbf{X})^2) + \frac{\varepsilon \sigma^2}{2} \right)$$

We now give the proof of Proposition C.3

*Proof of Proposition C.3.* We now show that $\mathcal{F}_k$ admits a $\mathcal{D}$-discretization with threshold functions $\kappa_0(\delta) = 2 \cdot 8000^2 \ln(e/\delta)$, $\varepsilon_0 = 1$ and size function $N(\mathcal{D}, \varepsilon, m) = 8 \left( \frac{72 \cdot 10^4 \cdot 8000 e}{\varepsilon} \right)^{140kd \ln(6k)}$ if $\sigma^2 = \mathbb{E}_{\mathbf{X} \sim \mathcal{D}}[d(\mathbf{X}, \mu)^2] < \infty$. Thus, for any $0 < \delta, \varepsilon < 1$ we get by invoking Theorem 3.4 that for $v_p \geq \sup_{f \in \mathcal{F}_k} \mathbb{E}_{\mathbf{X} \sim \mathcal{D}}[|f(\mathbf{X}) - \mathbb{E}_{\mathbf{X} \sim \mathcal{D}}[f(\mathbf{X})]|^p]$, $m \geq \left( \frac{400 \cdot 16^p v_p}{\varepsilon^p} \right)^{\frac{1}{p-1}}$ and $\kappa \geq \kappa_0(\delta/8), \frac{10^6 \ln(2)}{99}, 50 \ln \left( \frac{8N(\mathcal{D}, \varepsilon/16, m)}{\delta} \right)$, it holds with probability at least $1 - \delta$ over $\mathbf{X} \sim (\mathcal{D}^m)^\kappa$ that: For all $f \in \mathcal{F}_k$

$$| \text{MoM}(f, \mathbf{X}) - \mathbb{E}_{\mathbf{X}' \sim \mathcal{D}}[f(\mathbf{X}')]| \leq \varepsilon,$$

which would conclude the proof.

To the end of showing that $\mathcal{F}_k$ admits the above claimed $\mathcal{D}$-discretization, let $0 < \varepsilon < \varepsilon_0 = 1$, $0 < \delta < 1$, $m \geq 1$ and $\kappa \geq \kappa_0(\delta)$. Now for each $i \in \{1, \ldots, \kappa\}$, let $G_i$ be the event

$$G_i = \left\{ \sum_{j=0}^2 \sum_{l=1}^m \frac{s(\mathbf{X}_{j,l}^i)}{3m} \leq 12 \cdot 8000 \right\}. \tag{29}$$

Now by Markov's inequality $\sigma^2 < \infty$ and that $s(x) = \frac{4d(x, \mu)^2}{\sigma^2} + 8$ it follows that

$$\mathbb{P}_{\mathbf{X}_0^i, \mathbf{X}_1^i, \mathbf{X}_2^i \sim \mathcal{D}^m} \left[ G_i^C \right] \leq \mathbb{E}_{\mathbf{X}_0^i, \mathbf{X}_1^i, \mathbf{X}_2^i \sim \mathcal{D}^m} \left[ \sum_{j=0}^2 \sum_{l=1}^m \frac{s(\mathbf{X}_{j,l}^i)}{3m} \frac{1}{12 \cdot 8000} \right] \leq \frac{12}{12 \cdot 8000} = \frac{1}{8000}, \tag{30}$$

which implies that $\mathbb{P}_{\mathbf{X}_0^i, \mathbf{X}_1^i, \mathbf{X}_2^i \sim \mathcal{D}^m}[G_i] \geq 1 - \frac{1}{8000}$. Now since the random variables $\mathbf{X}_0^i, \mathbf{X}_1^i, \mathbf{X}_2^i$ for $i \in \{1, \ldots, \kappa\}$ are independent we get that the events $G_i$ are independent, and it follows by an application of the multiplicative Chernoff bound, and $\kappa \geq \kappa_0(\delta) = 2 \cdot 8000^2 \ln(e/\delta)$ that

$$\mathbb{P}_{\mathbf{X}_0, \mathbf{X}_1, \mathbf{X}_2 \sim (\mathcal{D}^m)^\kappa} \left[ \sum_i^\kappa \mathbb{1}\{G_i\} \leq (1 - \frac{1}{8000}) \mathbb{E} \left[ \sum_i^\kappa \mathbb{1}\{G_i\} \right] \right] \leq \exp \left( -\frac{\kappa}{2 \cdot 8000^2} \right) \leq \delta. \tag{31}$$

Thus, we conclude that with probability at least $1 - \delta$ over $\mathbf{X}_0, \mathbf{X}_1, \mathbf{X}_2$, it holds that $I = \{i : i \in [\kappa], \mathbb{1}\{G_i\} = 1\}$ is such that $|I| \geq (1 - \frac{1}{8000}) \mathbb{E}[\sum_i^\kappa \mathbb{1}\{G_i\}] \geq (1 - \frac{1}{8000})^2 \kappa$, where the last inequality follows from Equation (30). Now consider any realization $X_0, X_1, X_2$ of $\mathbf{X}_0, \mathbf{X}_1, \mathbf{X}_2$ such that $|I| \geq (1 - \frac{1}{8000})^2 \kappa$. Now by the definition of $G_i$ in Equation (29) we have that

$$\sum_{i \in I} \sum_{j=0}^2 \sum_{l=1}^m \frac{s(X_{j,l}^i)}{3m|I|} \leq 12 \cdot 8000. \tag{32}$$

Now let $X_G$ denote the points set $\cup_{i \in I} X^i = \left\{ x | \exists i \in I, \exists j \in \{0, 1, 2\}, \exists l \in [m] \text{ such } x = X_{j,l}^i \right\}$, so with out multiplicity, and define the distribution $\mathcal{D}_G$ on $X_G$, where $\mathcal{D}_G(x) = \sum_{i \in I} \sum_{j=0}^2 \sum_{l=1}^m \frac{\mathbb{1}\{X_{j,l}^i = x\}}{3m|I|}$, so assigning points in $X_G$ weight

after their multiplicity in $X^i$ for $i \in I$. We notice that with this distribution we have that

$$\underset{\mathbf{X} \sim \mathcal{D}_G}{\mathbb{E}} [s(\mathbf{X})] = \sum_{x \in X_G} s(x) \mathcal{D}_G(x) = \sum_{x \in X_G} \sum_{i \in I} \sum_{j=0}^{2} \sum_{l=1}^{m} s(x) \frac{\mathbb{1}\left\{X_{j,l}^i = x\right\}}{3m|I|}$$

$$= \sum_{i \in I} \sum_{j=0}^{2} \sum_{l=1}^{m} \frac{s(X_{j,l}^i)}{3m|I|} \leq 12 \cdot 8000, \tag{33}$$

where the last inequality follows by Equation (32).

We now invoke Lemma C.2 to get an upper bound on the size of a maximal $\frac{\varepsilon}{3 \cdot 10^4}$-packing of $\mathcal{F}_k(X_G)$ with respect to $L_1(\mathcal{D}_G)$ of size at most $8\left(\frac{6 \cdot 10^4 e \, \mathbb{E}_{\mathbf{X} \sim \mathcal{D}_G}[s(\mathbf{x})]}{\varepsilon}\right)^{2d_k}$ with $d_k = Pdim(\mathcal{F}_k)$. We here make a small remark about the invocation of Lemma C.2. Our $s$ is only an upper bound on $s'(x) = \sup_{f \in \mathcal{F}_k} f(x)$ (by Lemma C.1), thus we actually invoke Lemma C.2 with $s'$ and get that for $\varepsilon' = \min(\frac{\varepsilon}{3 \cdot 10^4}, \mathbb{E}_{\mathbf{X} \sim \mathcal{D}_G}[s'(\mathbf{X})])$ the size of a maximal $\varepsilon'$-packing is at most $8\left(\frac{2e \, \mathbb{E}_{\mathbf{X} \sim \mathcal{D}_G}[s'(x)]}{\varepsilon'}\right)^{2d_k}$.[1] We notice that if $\varepsilon' = \mathbb{E}_{\mathbf{X} \sim \mathcal{D}_G}[s'(\mathbf{X})]$, then since $\mathbb{E}_{\mathbf{X} \sim \mathcal{D}_G}[s(\mathbf{X})] \geq 4$ by Lemma C.1 and $0 < \varepsilon < 1$, it holds that $8\left(\frac{2 \, \mathbb{E}_{\mathbf{X} \sim \mathcal{D}_G}[s'(\mathbf{X})]}{\varepsilon'}\right)^{2d_k} \leq 8\left(\frac{6 \cdot 10^4 \, \mathbb{E}_{\mathbf{X} \sim \mathcal{D}_G}[s(\mathbf{X})]}{\varepsilon}\right)^{2d_k}$, if $\varepsilon' = \frac{\varepsilon}{3 \cdot 10^4}$, the above is also an upper bound since $s' \leq s$. Furthermore, the size of a maximal $\varepsilon'$-packing of $\mathcal{F}_k$ in $L_1(\mathcal{D}_G)$ is larger than the size of a minimal $\varepsilon'$-net of $\mathcal{F}_k$ in $L_1(\mathcal{D}_G)$. Thus, the above also gives that the size of a minimal $\frac{\varepsilon}{3 \cdot 10^4}$-net of $\mathcal{F}_k$ in $L_1(\mathcal{D}_G)$ is at most $8\left(\frac{6 \cdot 10^4 e \, \mathbb{E}_{\mathbf{X} \sim \mathcal{D}_G}[s(\mathbf{x})]}{\varepsilon}\right)^{2d_k}$. By Lemma C.1, we have that $d_k \leq 6k(d+4) \ln(6k)/\ln(2) \leq 70kd \ln(6k)$. By Equation (33) we have that $\mathbb{E}_{\mathbf{X} \sim \mathcal{D}_G}[s(\mathbf{x})] \leq 12 \cdot 8000$. Thus, we have argued that the size of a minimal $\frac{\varepsilon}{3 \cdot 10^4}$-net of $\mathcal{F}_k(X_G)$ in terms of $L_1(\mathcal{D}_G)$ is at most $8\left(\frac{6 \cdot 10^4 e \, \mathbb{E}_{\mathbf{X} \sim \mathcal{D}_G}[s(\mathbf{x})]}{\varepsilon}\right)^{2d_k} \leq 8\left(\frac{72 \cdot 10^4 \cdot 8000e}{\varepsilon}\right)^{140kd \ln(6k)} = N(\mathcal{D}, \varepsilon, m)$. Let $F_\varepsilon = F_\varepsilon(X_G, L_1, \mathcal{D}_G)$ denote a minimal $\frac{\varepsilon}{3 \cdot 10^4}$-net for $\mathcal{F}_k(X_G)$ with respect to $L_1(\mathcal{D}_G)$, i.e. has the following property, for $f \in \mathcal{F}_k(X_G)$, there $\exists f' \in F_\varepsilon$ such that

$$\frac{\varepsilon}{3 \cdot 10^4} \geq \underset{\mathbf{X} \sim \mathcal{D}_G}{\mathbb{E}}[|f(\mathbf{X}) - f'(\mathbf{X})|] = \sum_{i \in I} \sum_{j=0}^{2} \sum_{l=1}^{m} \frac{|f(X_{j,l}^i) - f'(X_{j,l}^i)|}{3m|I|}, \tag{34}$$

and any other set with this property has a size less than or equal to $|F_\varepsilon|$ - the above last equality follows by similar calculations as in Equation (33) (using the definition of $\mathcal{D}_G$). In what follows we will for $f \in \mathcal{F}_k$ use $\pi(f)$ for the element in $F_\varepsilon$ closest to $f$ with ties broken arbitrarily.

Let now $f \in \mathcal{F}_k$. We define $I_f$ to be the following subset of $[\kappa]$,

$$I_f = \left\{i \middle| \exists i \in [\kappa], \exists j \in \{0,1,2\} \sum_{l=1}^{m} \frac{|f(X_{j,l}^i) - \pi(f)(X_{j,l}^i)|}{m} > \varepsilon\right\}. \tag{35}$$

We first notice that by Equation (34) we have that

$$\frac{\varepsilon}{3 \cdot 10^4} \geq \sum_{i \in I \cap I_f} \sum_{j=0}^{2} \sum_{l=1}^{m} \frac{|f(X_{j,l}^i) - f'(X_{j,l}^i)|}{3m|I|} \geq \frac{|I \cap I_f| \varepsilon}{3|I|}$$

which implies that $|I \cap I_f| \leq \frac{1}{10^4}|I| \leq \frac{\kappa}{10^4}$. Furthermore since $|I| \geq (1 - \frac{1}{8000})^2 \kappa$, implying $I^C \leq (1 - (1 - \frac{1}{8000})^2)\kappa$, we conclude that

$$|I_f| = |I_f \cap I| + |I_f \cap I^C| \leq \left(\frac{1}{10^4} + 1 - (1 - \frac{1}{8000})^2\right)\kappa \leq \frac{2}{625}\kappa.$$

Thus, we have shown that for the realization $X_0, X_1, X_2$ there exists a set of functions $F_\varepsilon$ defined on $X_0, X_1, X_2$ such that for $f \in \mathcal{F}_k$ there exists $\pi(f) \in F_\varepsilon$ and $I_f$ such that $|I_f| \leq \frac{2}{625}\kappa$ and for $i \in [k] \backslash I_f$, and $j \in \{0,1,2\}$ we have that (see

---

[1] In the case that $\mathbb{E}_{\mathbf{X} \sim \mathcal{D}_G}[s'(\mathbf{X})] = 0$ we have $s'(x) = 0$ for $x \in X_G$, and we can take the cover to only consist of the 0 function on $X_G$, as $s'(x) \geq f(x) \geq 0$ so must be zero on $X_G$, thus we assume that $e_{\mathbf{X} \sim \mathcal{D}_G} > 0$.

Equation (35)) that

$$\sum_{l=1}^{m} \frac{|f(X_{j,l}^i) - \pi(f)(X_{j,l}^i)|}{m} \leq \varepsilon$$

and $|F_\varepsilon| \leq 8 \left( \frac{72 \cdot 10^4 \cdot 8000e}{\varepsilon} \right)^{140kd \ln (6k)} = N(\mathcal{D}, \varepsilon, m)$ (see the argument above Equation (34)). That is $\mathcal{F}_k$ admits a $\varepsilon$-discretization on $X_0, X_1, X_2$ of size at most $|F_\varepsilon| \leq 8 \left( \frac{72 \cdot 10^4 \cdot 8000e}{\varepsilon} \right)^{140kd \ln (6k)} = N(\mathcal{D}, \varepsilon, m)$, for $0 < \varepsilon \leq 1 = \varepsilon_0$. We showed the above for a realization $X_0, X_1, X_2$ such that $\mathbf{X}_0, \mathbf{X}_1, \mathbf{X}_2$ such that $|I| \geq (1 - \frac{1}{8000})^2 \kappa$, which happens with probability at least $1 - \delta$ for $\kappa \geq \kappa_0(\delta) = 2 \cdot 8000^2 \ln (e/\delta)$ and $\sigma^2 < \infty$ by Equation (31). Thus, we have shown that $\mathcal{F}_k$ admits the claimed $\mathcal{D}$-discretization which concludes the proof of Proposition C.3. □

## D. Linear Regression

In this appendix, we consider a continuous positive loss function $c_l \in [0, \infty)^{\mathbb{R}}$ and the function class induced by the loss function $\ell$ when doing linear regression with a constraint on the norm of the regressor of $W > 0$, formally we consider the function class

$$\mathcal{F}_W = \left\{ \ell(\langle (w, -1), \cdot \rangle) \mid w \in \mathbb{R}^d, ||w|| \leq W \right\} \subseteq [0, \infty)^{\mathbb{R}^{d+1}}.$$

That is if $f \in \mathcal{F}_W$ there exists $w \in \mathbb{R}^d$, with squared norm at most $W$ such that for $x \in \mathbb{R}^d$, $y \in \mathbb{R}$, we have $f((x, y)) = \ell(\langle (w, -1), (x, y) \rangle) = \ell(\langle w, x \rangle - y)$, i.e. the loss function $\ell$ taken on the residual. To describe our result we for a continuous loss function $\ell$ define for $a, b > 0$, the number $\alpha_\ell(a, b)$ as the largest positive number such that for $x, y \in [-a, a]$ and $|x - y| \leq \alpha_\ell(a, b)$ we have that $|\ell(x) - \ell(y)| \leq b$. We notice that since $\ell$ is continuous and $[a, a]$ is a compact interval $\alpha_\ell(a, b)$ is well-defined by Heine-Cantor Theorem which ensures that $\ell$ is uniform continuous on $[a, a]$. We notice that if $\ell$ is $L$-Lipschitz then we have that $\alpha_\ell(a, b) = \frac{b}{L}$.

In what follows we are going to need the following lemma which gives a bound on an epsilon net of the units ball in $\mathbb{R}^d$ in terms of $||\cdot||_2$.

**Lemma D.1.** *Let $0 < \varepsilon < 1$, $\mathrm{B}(1) = \left\{ x \in \mathbb{R}^d, ||X||_2 \leq 1 \right\}$, then there exists a set $B_\varepsilon \subseteq \mathbb{R}^d$, of size at most $(6/\varepsilon)^d$ being a $\varepsilon$-net for $\mathrm{B}(1)$ in $||\cdot||_2$, i.e. for any $x \in \mathrm{B}(1)$ there exists $y \in B_\varepsilon$ such that*

$$||x - y||_2 \leq \varepsilon. \tag{36}$$

We postpone the proof of Lemma D.1 to the end of this appendix and now present our main proposition on regression for a continuous loss function.

**Proposition D.2** (Regression). *For a continuous loss function $\ell$, $0 < \delta < 1$, $1 < p \leq 2$, distributions $\mathcal{D}_X$ and $\mathcal{D}_Y$ over respectively $\mathbb{R}^d$ and $\mathbb{R}$, $W > 0$, $\infty > v_p \geq \sup_{f \in \mathcal{F}_W} \mathbb{E}_{\mathbf{X} \sim \mathcal{D}_X, \mathbf{Y} \sim \mathcal{D}_Y} [f(\mathbf{X}, \mathbf{Y})^p]$, $\kappa_0(\delta) = 4 \cdot 1250^2 \ln (e/\delta)$, and $N(\mathcal{D}, \varepsilon, m) = \left( \frac{6W}{\beta(\varepsilon, m, \mathcal{D})} \right)^d$ where $\beta(\varepsilon, m, \mathcal{D}) = \min(\frac{W}{2}, \frac{\alpha_\ell(J, \varepsilon)}{3750(\mathbb{E}[||X||_1] + \mathbb{E}[|Y|])m})$, with $J = (3W/2 + 1) \cdot 3750 (\mathbb{E}[||X||_1] + \mathbb{E}[|Y|]) m$ we then have that for $0 < \varepsilon$, $m \geq \left( \frac{400 \cdot 16^p v_p}{\varepsilon^p} \right)^{\frac{1}{p-1}}$ and $\kappa \geq \max \left( \kappa_0(\delta/8), \frac{10^6 \ln (2)}{99}, 50 \ln \left( \frac{8N(\mathcal{D}, \varepsilon/16, m)}{\delta} \right) \right)$, it holds with probability at least $1 - \delta$ over $\mathbf{Z} = (\mathbf{X}, \mathbf{Y}) \sim ((\mathcal{D}_X \times \mathcal{D}_Y)^m)^\kappa$ that: For all $f \in \mathcal{F}_W$*

$$| \mathrm{MoM}(f, \mathbf{Z}) - \mathbb{E}_{\mathbf{Z}' \sim \mathcal{D}_X \times \mathcal{D}_Y} [f(\mathbf{Z}')]| \leq \varepsilon.$$

Before we prove Proposition D.2 we make a small remark on what Proposition D.2 means for Lipschitz losses. In the case

of $\ell$ being an $L$-Lipschitz loss and that $\ell(0) = 0$ we get that

$$\sup_{f\in\mathcal{F}_{W,\ell}} \mathbb{E}\left[f(\mathbf{X},\mathbf{Y})^p\right] \leq L^p \sup_{w\in\mathrm{B}(W)} \mathbb{E}\left[|\langle(w,-1),(\mathbf{X},\mathbf{Y})\rangle|^p\right] \qquad \text{(by } \ell(0) = 0 \text{ and } \ell \text{ L-Lipschitz)}$$

$$\leq L^p \sup_{w\in\mathrm{B}(W)} \mathbb{E}\left[(|\langle w,\mathbf{X}\rangle| + |\mathbf{Y}|)^p\right] \qquad \text{(by } |a+b| \leq |a| + |b|)$$

$$\leq (2L)^p \sup_{w\in\mathrm{B}(W)} \mathbb{E}\left[(\max\left(|\langle w,\mathbf{X}\rangle|^p, |\mathbf{Y}|^p\right))\right] \qquad \text{(by } a+b \leq 2\max(a,b))$$

$$\leq (2L)^p \left(\sup_{w\in\mathrm{B}(W)} \mathbb{E}\left[|\langle w,\mathbf{X}\rangle|^p\right] + \mathbb{E}\left[|\mathbf{Y}|^p\right]\right). \qquad \text{(by } \max(a,b) \leq a+b)$$

Furthermore as discussed earlier we in this case have that $\alpha_\ell(J, \varepsilon/16) = \frac{\varepsilon}{16L}$, plugging these observations into Proposition D.2 implies the following corollary.

**Corollary D.3** (Lipschitz-loss). *For a continuous loss function $\ell$ which is $L - Lipschitz$ and has $\ell(0) = 0$, $0 < \delta < 1$, $1 < p \leq 2$, distributions $\mathcal{D}_X$ and $\mathcal{D}_Y$ over respectively $\mathbb{R}^d$ and $\mathbb{R}$, $W > 0$, $\infty > v_p \geq (2L)^p \left(\sup_{w\in\mathrm{B}(W)} \mathbb{E}\left[|\langle w,\mathbf{X}\rangle|^p\right] + \mathbb{E}\left[|\mathbf{Y}|^p\right]\right)$, $\kappa_0(\delta) = 4 \cdot 1250^2 \ln(e/\delta)$, and $N(\mathcal{D}, \varepsilon, m) = \left(\frac{6W}{\beta(\varepsilon,m,\mathcal{D})}\right)^d$ where $\beta(\varepsilon, m, \mathcal{D}) = \min(\frac{W}{2}, \frac{\varepsilon}{60000L(\mathbb{E}[||X||_1]+\mathbb{E}[|Y|])m})$, we have that for $0 < \varepsilon$, $m \geq \left(\frac{400\cdot16^p v_p}{\varepsilon^p}\right)^{\frac{1}{p-1}}$ and $\kappa \geq \max\left(\kappa_0(\delta/8), \frac{10^6 \ln(2)}{99}, 50\ln\left(\frac{8N(\mathcal{D},\varepsilon/16,m)}{\delta}\right)\right)$, it holds with probability at least $1 - \delta$ over $\mathbf{Z} = (\mathbf{X}, \mathbf{Y}) \sim ((\mathcal{D}_X \times \mathcal{D}_Y)^m)^\kappa$ that: For all $f \in \mathcal{F}_W$*

$$\left|\mathrm{MoM}(f,\mathbf{Z}) - \mathbb{E}_{\mathbf{Z}'\sim\mathcal{D}_X\times\mathcal{D}_Y}[f(\mathbf{Z}')]\right| \leq \varepsilon.$$

We now give the proof of Proposition D.2

*Proof of Proposition D.2.* In what follows we will use $\mathbf{Z}_0 = (\mathbf{X}_0, \mathbf{Y}_0)$, $\mathbf{Z}_1 = (\mathbf{X}_1, \mathbf{Y}_1)$, and $\mathbf{Z}_2 = (\mathbf{X}_2, \mathbf{Y}_2)$ and let $\mathcal{D}_Z = \mathcal{D}_X \times \mathcal{D}_Y$, such that $\mathbf{Z}_l \sim (\mathcal{D}_Z^m)^\kappa$. We now show that $\mathcal{F}_W$ admits a $\mathcal{D}$-discretization with threshold functions $\kappa_0(\delta) = 4 \cdot 1250^2 \ln(e/\delta)$, $\varepsilon_0 = \infty$ and size function $N(\mathcal{D}, \varepsilon, m) = \left(\frac{6W}{\beta(\varepsilon,m,\mathcal{D})}\right)^d$ where $\beta(\varepsilon, m, \mathcal{D}) = \min(\frac{W}{2}, \frac{\alpha(J,\varepsilon)}{3750(\mathbb{E}[||X||_1]+\mathbb{E}[|Y|])m})$, with $J = (3W/2 + 1) \cdot 3750\left(\mathbb{E}\left[||X||_1\right] + \mathbb{E}\left[|Y|\right]\right)m$. Thus, for any $0 < \delta < 1$ we get by invoking Theorem 3.4 that for $0 < \varepsilon < \varepsilon_0 = \infty$, $m \geq \left(\frac{400\cdot16^p v_p}{\varepsilon^p}\right)^{\frac{1}{p-1}}$ and $\kappa \geq \kappa_0(\delta/8), \frac{10^6 \ln(2)}{99}, 50\ln\left(\frac{8N(\mathcal{D},\varepsilon/16,m)}{\delta}\right)$ it holds with probability at least $1 - \delta$ over $\mathbf{Z} \sim (\mathcal{D}_Z^m)^\kappa$ that: For all $f \in \mathcal{F}$

$$\left|\mathrm{MoM}(f,\mathbf{Z}) - \mathbb{E}_{\mathbf{Z}'\sim\mathcal{D}_Z}[f(\mathbf{Z}')]\right| \leq \varepsilon,$$

as claimed. To the end of showing that $\mathcal{F}_W$ admits a $\mathcal{D}$-discretization as described above let $0 < \varepsilon < \varepsilon_0 = \infty$, $0 < \delta < 1$, $m \geq 1$ and $\kappa \geq \kappa_0(\delta)$. We now define the following events for $i \in [\kappa]$

$$G_i = \left\{\sum_{l=0}^{2}\sum_{j=1}^{m} \frac{\left|\left|(\mathbf{X}_{l,j}^i, \mathbf{Y}_{l,j}^i)\right|\right|_1}{3m} \leq 1250\left(\mathbb{E}\left[||X||_1\right] + \mathbb{E}\left[|Y|\right]\right)\right\}.$$

Then by Markov's inequality, we have that

$$\mathbb{P}_{\mathbf{Z}_0^i,\mathbf{Z}_1^i,\mathbf{Z}_2^i\sim\mathcal{D}_Z^m}\left(G_i^C\right) \leq \mathbb{E}\left[\sum_{l=0}^{2}\sum_{j=1}^{m} \frac{\left|\left|(\mathbf{X}_{l,j}^i, \mathbf{Y}_{l,j}^i)\right|\right|_1}{3m} \frac{1}{1250\left(\mathbb{E}\left[||X||_1\right] + \mathbb{E}\left[|Y|\right]\right)}\right] = \frac{1}{1250},$$

which implies that $\mathbb{P}_{\mathbf{Z}_0^i,\mathbf{Z}_1^i,\mathbf{Z}_2^i\sim\mathcal{D}_Z^m}[G_i] \geq 1 - \frac{1}{1250}$. Now since the random variables $\mathbf{Z}_0^i, \mathbf{Z}_1^i, \mathbf{Z}_2^i$ for $i \in \{1,\ldots,\kappa\}$ are independent we get that the events $G_i$ are independent. Thus, it follows by an application of the multiplicative Chernoff bound, that for any $\kappa \geq \kappa_0(\delta) = 4 \cdot 1250^2 \ln(e/\delta)$ we have

$$\mathbb{P}_{\mathbf{Z}_0,\mathbf{Z}_1,\mathbf{Z}_2\sim(\mathcal{D}^m)^\kappa}\left[\sum_{i}^{\kappa} \mathbb{1}\{G_i\} \leq (1 - \frac{1}{1250})\mathbb{E}\left[\sum_{i}^{\kappa} \mathbb{1}\{G_i\}\right]\right] \leq \exp\left(-\frac{(1-\frac{1}{1250})\kappa}{2\cdot1250^2}\right) \leq \delta. \qquad (37)$$

Thus, if we let $I = \{i \in [\kappa] : \mathbb{1}\{G_i\} = 1\}$ then by Equation (37) it holds with probability at least $1 - \delta$ over $\mathbf{Z}_0, \mathbf{Z}_1, \mathbf{Z}_2$ that $|I| \geq (1 - \frac{1}{1250})^2 \kappa$, where we have used that we concluded earlier that $\mathbb{P}_{\mathbf{Z}_0^i, \mathbf{Z}_1^i, \mathbf{Z}_2^i \sim \mathcal{D}_Z^m}[G_i] \geq 1 - \frac{1}{1250}$. Thus, we have that the size $I^C = [\kappa] \backslash I$, is at most $|I^C| = \kappa - |I| \leq \kappa - (1 - \frac{1}{1250})^2 \kappa \leq \frac{2}{625} \kappa$. Thus, if for $Z_0, Z_1, Z_2$, outcomes of $\mathbf{Z}_0, \mathbf{Z}_1, \mathbf{Z}_2$ where $|I| \geq (1 - \frac{1}{1250})^2 \kappa$, we can construct a set of functions $F_\beta$ ($\beta$ is a parameter depending on $\varepsilon, m, \mathcal{D}$ as allowed to in the definition of a $\mathcal{D}$-discretization ) defined on $Z_0, Z_1, Z_2$ such that for $f \in \mathcal{F}_W$, there exists $\pi(f) \in F_\beta$, such that for $i \in [\kappa] \backslash I^C = I$ it holds that for each $l \in \{0, 1, 2\}$ that

$$\sum_{j=1}^{m} \left| \frac{f(Z_{l,j}^i) - \pi(f)(Z_{l,j}^i)}{m} \right| \leq \varepsilon, \tag{38}$$

and the size of $|F_\beta| \leq N(\mathcal{D}, \varepsilon, m)$ then we have shown that $\mathcal{F}_W$ admits a $\mathcal{D}$ discretization, and we are done by the above. Thus, we now show that for outcomes $Z_0, Z_1, Z_2$, of $\mathbf{Z}_0, \mathbf{Z}_1, \mathbf{Z}_2$ where $|I| \geq (1 - \frac{1}{1250})^2 \kappa$, there exists such an $F_\beta$.

To this end consider a realization $Z_0, Z_1, Z_2$ of $\mathbf{Z}_0, \mathbf{Z}_1, \mathbf{Z}_2$ such that $|I| \geq (1 - \frac{1}{1250})^2 \kappa$. We first notice that for any $i \in I$, $l \in \{0, 1, 2\}$ we have that

$$\sum_{j=1}^{m} \frac{\left\| (X_{l,j}^i, Y_{l,j}^i) \right\|_1}{3m} \leq 1250 \left( \mathbb{E}\left[ \|X\|_1 \right] + \mathbb{E}\left[ |Y| \right] \right),$$

which implies that for any $j \in [m]$

$$\left\| (X_{l,j}^i, Y_{l,j}^i) \right\|_1 \leq 3750 \left( \mathbb{E}\left[ \|X\|_1 \right] + \mathbb{E}\left[ |Y| \right] \right) m.$$

Thus, we conclude that for any $i \in I$, $l \in \{0, 1, 2\}$ and $j \in [m]$ we have that

$$\left\| (X_{l,j}^i, Y_{l,j}^i) \right\|_1 \leq 3750 \left( \mathbb{E}\left[ \|X\|_1 \right] + \mathbb{E}\left[ |Y| \right] \right) m. \tag{39}$$

Now let $\beta = \beta(\varepsilon, m, \mathcal{D}) = \min(\frac{W}{2}, \frac{\alpha(J, \varepsilon)}{3750(\mathbb{E}[\|X\|_1] + \mathbb{E}[|Y|])m})$, where $J$ denote the quantity $J = (3W/2 + 1) \cdot 3750 \left( \mathbb{E}\left[ \|X\|_1 \right] + \mathbb{E}\left[ |Y| \right] \right) m$ and let $F_\beta$ denote a $\beta$-net in $\|\cdot\|_2$-norm for $\mathrm{B}(W) = \{w \in \mathbb{R}^d : \|w\|_2 \leq W\}$ of minimal size, i.e. $\forall w \in \mathrm{B}(W)$ there $\exists \hat{w} \in F_\beta$ such that $\|w - \hat{w}\|_2 \leq \beta$, and any other set satisfying this has size at least $|F_\beta|$. We notice that this implies that for $\hat{w} \in F_\beta$ we have that $\|\hat{w}\| \leq W + \beta$, to see this let $w \in \mathrm{B}(W)$ and let $\hat{w} \in F_\beta$ be the points closest to $w$ in the net $F_\beta$, i.e. we have that $\|w - \hat{w}\| \leq \beta$, and by the reverse triangle inequality we have that $W \geq \|w\| \geq \|\hat{w}\|_2 - \|\hat{w} - w\|_2 > \|\hat{w}\|_2 - \beta$ which implies that $\|\hat{w}\|_2 \leq W + \beta$ as claimed. Now using Equation (39), that $\|\cdot\|_2 \leq \|\cdot\|_1$ and Cauchy Schwarz it follows that for $w \in \mathrm{B}(W)$, $\hat{w}$ the points closest to $w$ from $F_\beta$(ties broken arbitrarily), $i \in I$, $l \in \{0, 1, 2\}$ and $j \in [m]$ that

$$|\langle (w, -1), (X_{l,j}^i, Y_{l,j}^i) \rangle - \langle (\hat{w}, -1), (X_{l,j}^i, Y_{l,j}^i) \rangle | = |\langle (w, -1) - (\hat{w}, -1), (X_{l,j}^i, Y_{l,j}^i) \rangle | \tag{40}$$
$$\leq \beta \cdot \left\| (X_{l,j}^i, Y_{l,j}^i) \right\|_2 \qquad \text{(by Cauchy Schwarz)}$$
$$\leq \beta \cdot \left\| (X_{l,j}^i, Y_{l,j}^i) \right\|_1 \qquad \text{(by } \|\cdot\|_2 \leq \|\cdot\|_1 )$$
$$\leq \beta \cdot 3750 \left( \mathbb{E}\left[ \|X\|_1 \right] + \mathbb{E}\left[ |Y| \right] \right) m \qquad \text{(by Equation (39))}$$

and that for any $w' \in \mathrm{B}(W) \cup F_\beta$, $i \in I$, $l \in \{0, 1, 2\}$ and $j \in [m]$ that

$$|\langle (w', -1), (X_{l,j}^i, Y_{l,j}^i) \rangle | \leq \|(w', -1)\|_2 \left\| (X_{l,j}^i, Y_{l,j}^i) \right\|_2 \qquad \text{(by Cauchy Schwarz)}$$
$$\leq (\|(w')\|_2 + 1) \left\| (X_{l,j}^i, Y_{l,j}^i) \right\|_2 \qquad \text{(by } \sqrt{a+b} \leq \sqrt{a} + \sqrt{b})$$
$$\leq (W + \beta + 1) \left\| (X_{l,j}^i, Y_{l,j}^i) \right\|_2 \qquad \text{(by } w' \in \mathrm{B}(W) \cup F_\beta \text{ so } \|w\|_2 \leq W + \beta)$$
$$\leq (W + \beta + 1) \left\| (X_{l,j}^i, Y_{l,j}^i) \right\|_1 \qquad \text{(by } \|\cdot\|_2 \leq \|\cdot\|_1 )$$
$$\leq (W + \beta + 1) \cdot 3750 \left( \mathbb{E}\left[ \|X\|_1 \right] + \mathbb{E}\left[ |Y| \right] \right) m \qquad \text{(by Equation (39))}$$
$$\leq (3W/2 + 1) \cdot 3750 \left( \mathbb{E}\left[ \|X\|_1 \right] + \mathbb{E}\left[ |Y| \right] \right) m \qquad \text{(by } \beta \leq \frac{W}{2})$$

Let now $J = (3W/2 + 1) \cdot 3750 \left( \mathbb{E}\left[ \|X\|_1 \right] + \mathbb{E}\left[ |Y| \right] \right) m$ and consider $\alpha_\ell(J, \varepsilon)$. We recall that by the definition of $\alpha_\ell(J, \varepsilon)$ we have that for $x, y \in [-J, J]$, such that $|x - y| \leq \alpha_\ell(J, \varepsilon)$ it holds that $|\ell(x) - \ell(y)| \leq \varepsilon$. Thus, by Equation (40) we

have that for any $i \in I$, $l \in \{0, 1, 2\}$ $j \in [m]$ and $w \in \mathrm{B}(W)$ with $\hat{w}$ being the point in $F_\beta$ closest to $w$ (with ties broken arbitrarily) that

$$|\langle (w, -1), (X^i_{l,j}, Y^i_{l,j}) \rangle - \langle (\hat{w}, -1), (X^i_{l,j}, Y^i_{l,j}) \rangle | \leq \beta \cdot 3750 \left( \mathbb{E}\left[ ||X||_1 \right] + \mathbb{E}\left[ |Y| \right] \right) m \leq \alpha_\ell(J, \varepsilon),$$

by $\beta = \min(\frac{W}{2}, \frac{\alpha(J,\varepsilon)}{3750(\mathbb{E}[||X||_1]+\mathbb{E}[|Y|])m})$, which implies that

$$|\ell(\langle (w, -1), (X^i_{l,j}, Y^i_{l,j}) \rangle) - \ell \left( \langle (\hat{w}, -1), (X^i_{l,j}, Y^i_{l,j}) \rangle \right) | \leq \varepsilon,$$

and furthermore for $i \in I$ and $l \in \{0, 1, 2\}$

$$\sum_{i=1}^m \left| \frac{\ell \left( \langle (w, -1), (X^i_{l,j}, Y^i_{l,j}) \rangle \right) - \ell \left( \langle (\hat{w}, -1), (X^i_{l,j}, Y^i_{l,j}) \rangle \right)}{m} \right| \leq \varepsilon,$$

which concludes Equation (38), i.e. that $F_\beta$ (formally speaking $\ell$ compossed with the vectors in $F_\beta$) is a $\varepsilon$-discretization of the realization $Z_0, Z_1, Z_2$ of $\mathbf{Z}_0, \mathbf{Z}_1, \mathbf{Z}_2$. Furthermore, by Lemma D.1 and $\beta/W \leq 1/2$ there exists a net $B_{\beta/W}$ of $\mathrm{B}(1)$ with precision $\beta/W$ in $||\cdot||_2$ of size at most $(6W/\beta)^d$. We now notice that for any $w \in \mathrm{B}(W)$, since $\frac{w}{W} \in \mathrm{B}(1)$ we have that there exists $\hat{w} \in B_{\beta/W}$ such that $\left\| \frac{w}{W} - \hat{w} \right\|_2 \leq \frac{\beta}{W}$ which implies that $||w - W\hat{w}||_2 \leq \beta$, i.e. shows that $W B_{\beta/W} = \{\hat{w} | \hat{w} = W w' \text{ for } w' \in B_{\beta/W}\}$, is a $\beta$ net for $\mathrm{B}(W)$ in $||\cdot||_2$. Thus, since $F_\beta$ was chosen minimally (in terms of size) over such nets, we have that the size of $|F_\beta| \leq \left( \frac{6W}{\beta} \right)^d$, where $\beta = \beta(\varepsilon, m, \mathcal{D}) = \min(\frac{W}{2}, \frac{\alpha_\ell(J,\varepsilon)}{3750(\mathbb{E}[||X||_1]+\mathbb{E}[|Y|])m})$, with $J = (3W/2 + 1) \cdot 3750 \left( \mathbb{E}\left[ ||X||_1 \right] + \mathbb{E}\left[ |Y| \right] \right) m$. Which concludes the claim of $\mathcal{F}_W$ having a $\mathcal{D}$-discretization with threshold functions $\varepsilon_0 = \infty$, $\kappa_0(\delta) = 4 \cdot 1250^2 \ln(e/\delta)$ and size function $N(\mathcal{D}, \varepsilon, m) = \left( \frac{6W}{\beta(\varepsilon,m,\mathcal{D})} \right)^d$ where $\beta(\varepsilon, m, \mathcal{D}) = \min(\frac{W}{2}, \frac{\alpha(J,\varepsilon)}{3750(\mathbb{E}[||X||_1]+\mathbb{E}[|Y|])m})$, with $J = (3W/2 + 1) \cdot 3750 \left( \mathbb{E}\left[ ||X||_1 \right] + \mathbb{E}\left[ |Y| \right] \right) m$ and further concludes the proof. $\qquad \square$

We now give the proof of Lemma D.1

*Proof of Lemma D.1.* In the following we are going to need that the volume of a ball in $\mathbb{R}^d$ of radius $r$ is $\mathrm{Vol}(B(r)) = \frac{\pi^{d/2}}{\Gamma(d/2+1)} r^d$ by 5.19(iii), where $\Gamma$ is the Euler's gamma function. It will also be convenient to introduce some notation for balls centered at a point $x \in \mathbb{R}^d$ of radius $r$, we will use $\mathrm{B}(x, r) = \{y | y \in \mathbb{R}^d, ||x - y||_2 \leq r\}$ for such balls.

Now let $B_\varepsilon$ be a maximal $\varepsilon$ packing of $\mathrm{B}(1)$, that is $B_\varepsilon \subseteq \mathrm{B}(1)$ and for any $x, y \in B_\varepsilon$, where $x \neq y$, we have that $||x - y||_2 \geq \varepsilon$, and any other subset of $\mathrm{B}(1)$ with this property has size less than or equal to $B_\varepsilon$. We notice that $B_\varepsilon$ must also be a $\varepsilon$-net for $\mathrm{B}(1)$ since else there exists $x \in \mathrm{B}(1)$ such that for all $y \in B_\varepsilon$ we have that $||x - y||_2 > \varepsilon$, but then $x$ could have been added to the maximal packing $B_\varepsilon$, leading to a contradiction with the maximality assumption of $B_\varepsilon$. We now argue that the size of $B_\varepsilon$ is $(6/\varepsilon)^d$, which would conclude the proof since we just argued it is a $\varepsilon$-net in $||\cdot||_2$ for $\mathrm{B}(1)$.

First since $B_\varepsilon$ is a packing it must be the case that if we place a ball on each point in $B_\varepsilon$ of radius $\varepsilon/3$, then these balls must be disjoint. To see this assume it is not the case i.e. there exists $x, y \in B_\varepsilon$ such that $x \neq y$ and the balls $\mathrm{B}(x, \varepsilon/3)$ and $\mathrm{B}(y, \varepsilon/3)$ centered at $x$ and $y$ of radius $\varepsilon/3$ has a nonempty intersection $\mathrm{B}(x, \varepsilon/3) \cap \mathrm{B}(y, \varepsilon/3) \neq \emptyset$. Now let $z$ be any point in this nonempty intersection, then by the triangle inequality we have that $||x - y||_2 \leq ||x - z||_2 - ||z - y||_2 \leq \frac{2\varepsilon}{3}$ leading to a contradiction with $B_\varepsilon$ being a $\varepsilon$-packing, i.e. all different elements being $\varepsilon$ away from each other.

Thus, we conclude that the balls centered at each point in $B_\varepsilon$ of radius $\varepsilon/3$ are disjoint, which implies that the sum of the volumes of the balls centered around each point in $B_\varepsilon$, which is $\sum_{x \in B_\varepsilon} \mathrm{Vol}(x, \varepsilon/3) = |B_\varepsilon| \mathrm{Vol}(B(\varepsilon/3))$, are equal to the volume of the union of all these balls $\mathrm{Vol}(\cup_{x \in B_\varepsilon} \mathrm{B}(x, \varepsilon/3)) = \sum_{x \in B_\varepsilon} \mathrm{Vol}(x, \varepsilon/3) = |B_\varepsilon| \mathrm{Vol}(B(\varepsilon/3))$.

We now notice that any point in a ball of radius $\varepsilon/3$ of a point in $B_\varepsilon$ has norm at most $1 + \varepsilon/3$. To see this let $x \in B_\varepsilon$ and $y \in \mathrm{B}(x, \varepsilon/3)$ be a point contained in the ball of radius $\varepsilon/3$ around $x$, then we have by the triangle inequality that $||y||_2 \leq ||x||_2 + ||y - x||_2 \leq 1 + \varepsilon/3$, where we in the last inequality used that $B_\varepsilon \subseteq \mathrm{B}(1)$ such that $||x||_2 \leq 1$. Thus, we have that the union of the balls of radius $\varepsilon/3$ centered at points in $B_\varepsilon$, $\cup_{x \in B_\varepsilon} \mathrm{B}(x, \varepsilon/3)$, is contained in the ball of radius $1 + \varepsilon/3$, $\cup_{x \in B_\varepsilon} \mathrm{B}(x, \varepsilon/3) \subseteq \mathrm{B}(1 + \varepsilon/3)$.

Thus, we conclude that $|B_\varepsilon| \mathrm{Vol}(B(\varepsilon/3)) = \mathrm{Vol}(\cup_{x \in B_\varepsilon} \mathrm{B}(x, \varepsilon/3)) \leq \mathrm{Vol}(B(1 + \varepsilon/3))$, where we used that we earlier conclude that the union of the balls of radius $\varepsilon/3$ centered at points in $B_\varepsilon$, $\cup_{x \in B_\varepsilon} \mathrm{B}(x, \varepsilon/3)$, have volume $|B_\varepsilon| \mathrm{Vol}(B(\varepsilon/3))$.

We notice that this implies combined with the formula for the volume of a ball in $\mathbb{R}^d$ of radius $r$ being $\text{Vol}(B(r)) = \frac{\pi^{d/2}}{\Gamma(d/2+1)} r^d$ that $|B_\varepsilon| \leq \frac{\text{Vol}(B(1+\varepsilon/3))}{\text{Vol}(B(\varepsilon/3))} = \left(\frac{1+\varepsilon/3}{\varepsilon/3}\right)^d \leq \left(\frac{6}{\varepsilon}\right)^d$, where the last inequality follows by $\varepsilon < 1$, and concludes the proof. $\qquad\square$

## E. Proofs of Lemma B.1.

**Lemma E.1** (Follows from (von Bahr & Esseen, 1965) Theorem 2). *Let $1 \leq p \leq 2$ and $\mathbf{X} = (\mathbf{X}_1, \ldots, \mathbf{X}_m)$ be i.i.d. with distribution $\mathcal{D}$. Furthermore let $\hat{\mu} = \frac{1}{m} \sum_{i=1}^{m} \mathbf{X}_i$, $\mu = \mathbb{E}_{\mathbf{X}_1 \sim \mathcal{D}}[\mathbf{X}_1]$ and $v_p \geq \mathbb{E}_{\mathbf{X}_1 \sim \mathcal{D}}[|\mathbf{X}_1 - \mu|^p]$. We then have that*

$$\mathbb{E}\left[|(\hat{\mu} - \mu)|^p\right] \leq \frac{2v_p}{m^{p-1}} \tag{41}$$

Using this lemma and Markovs inequality we obtain Lemma B.1.

*Proof of Lemma B.1.* By Lemma E.1 it holds for any $f \in \mathcal{F}$ that

$$\mathop{\mathbb{E}}_{\mathbf{X} \sim \mathcal{D}}[|\mu(f, \mathbf{X}) - \mu(f)|^p] \leq \frac{2v_p}{m^{p-1}}, \tag{42}$$

Using the lower bound of $m$ implies that we have that

$$\left(\frac{2v_p}{\delta m^{p-1}}\right)^{\frac{1}{p}} \leq \varepsilon,$$

thus by and Markovs inequality and Equation (42) we have that,

$$\mathbb{P}\left(|\mu(f, \mathbf{X}) - \mu(f)| > \varepsilon\right) \leq \mathbb{P}\left(|\mu(f, \mathbf{X}) - \mu(f)| > \left(\frac{2v_p}{\delta m^{p-1}}\right)^{\frac{1}{p}}\right)$$

$$\leq \frac{\mathbb{E}\left[|\mu(f, \mathbf{X}) - \mu(f)|^p\right]\delta m^{p-1}}{2v_p} \leq \delta,$$

which concludes the proof of Lemma B.1 $\qquad\square$

