# OpenReview forum: "Uniform Mean Estimation for Heavy-Tailed Distributions via Median-of-Means"
_ICML.cc/2025/Conference — ICML 2025 poster_

### Official Review · Reviewer_xSiA · 2025-02-19

**Overall Recommendation:** 3

**Summary:**

The paper examines the Median of Means (MoM) estimator for estimating means in heavy-tailed distributions. The authors derive a new sample complexity bound using an innovative symmetrization technique. They also present applications of this technique to k-means clustering with unbounded inputs and to linear regression with general loss functions. Overall, the study introduces a novel symmetrization method to achieve these results.

**Claims And Evidence:**

Yes, the authors provided theoretical to support their claims. However, I have some comments on some problematic claims:

The expressions in the main results—Theorem 3.4, Theorem 4.1, and Theorem 4.3—are not reader-friendly and can be difficult to follow. The meanings of \(\epsilon\) and \(\delta\) in the final formulas are quite ambiguous, and their relationship is not directly explained. This lack of clarity makes it challenging to understand the claims and complicates evaluation and comparison with previous works.

**Essential References Not Discussed:**

The literature review is extensive.

**Experimental Designs Or Analyses:**

This paper doesn't have experimental designs or analyses.

**Methods And Evaluation Criteria:**

1. No benchmark datasets are provided to demonstrate the application of the methods.

2. The paper does not clearly present the proposed method; for instance, there is no algorithm table showing each step of the process. This ambiguity makes application and evaluation very difficult.

**Other Comments Or Suggestions:**

1. Experimental analysis is preferred for readers.


=============================

Post rebuttal updates: Thank you for your detailed response to my reviews. I appreciate that all of my concerns have been addressed. I will increase my overall ratings based on the discussions in the rebuttal section, provided that my concerns and suggestions are included in the next version.

**Other Strengths And Weaknesses:**

**Other Strengths:**

1. The motivation of the paper is clear.

**Other Weaknesses:**

1. The assumptions of the theoretical analysis are not clearly stated.

2. The expression MOM in Theorem 3.4 is not well-defined; a clearer expression would be more reader-friendly.

**Questions For Authors:**

1.Is it possible to include some experimental analysis on the real data?

**Relation To Broader Scientific Literature:**

The ideas in this paper can be applied to k-means clustering with unbounded inputs and to linear regression with general losses, enhancing the existing approaches.

**Theoretical Claims:**

The notations in the theoretical proof are very ambiguous and contradictory, making it difficult to verify its correctness. For example, in Lemma 3.7 on page 5, the values are given as a = 4801/10000 and b = 9701/10000. However, in Lemma 3.8 on page 6, the values change to a = 4769/10000 and b = 331/10000. This inconsistency renders the proof problematic.

---

> ### Author Rebuttal · Authors · 2025-03-28
>
> We thank the reviewer for taking the time to read the manuscript and provide constructive feedback.
>
> In the following, we will address the reviewer’s comments in the order they were given, except for those regarding experiments, which we will address last.
>
> **Claims and Evidence:** We first address the reviewer’s comment on the main results being quite technical and difficult to follow. We will add further explanations in the paragraph following Theorem 3.4. Specifically, we will clarify that the precision parameter $\varepsilon$ affects both $m$ (the number of samples used in each mean estimate) and the precision of the net, with the intuition that we want both the mean estimates to be $O(\varepsilon)$-close to the mean and the "rounding" in the net to have precision $O(\varepsilon)$. Additionally, we will clarify that $\delta$ influences only the number of mean estimates $\kappa$, as this parameter is used to increase the probability that more than half of the $\kappa$ mean estimates are successful - this ensures that the median of the mean estimates, which is the output of MoM, is correct. We appreciate the reviewer’s suggestion to clarify this point.
>
> **Theoretical claims:** We agree that readability would be improved by keeping the constants in Lemma 3.8 as $c=\frac{4769}{10000}$ instead of $a=\frac{4769}{10000}$ and $d=\frac{331}{1000}$ instead of $b=\frac{331}{1000}$ to mirror the constants in Lemma 3.7. We will make this correction in the next version of the paper.
>
> **Methods And Evaluation Criteria:** We thank the review for pointing out that the estimator could be better presented. We will make an pseudo-algorithm environment illustrating the procedure of the MoM.
>
> **Weaknesses:** We thank the reviewer for pointing out that some of the theoretical assumptions could be more clearly stated. For example, in Theorem 3.4, we did not explicitly state that $\mathbf{X}=(\mathbf{X}\_1,\ldots,\mathbf{X}\_{\kappa})$ is drawn from $(\mathcal{D}^{m})^{\kappa}$, so that $\mathbf{X}\_i \sim \mathcal{D}^{m}$, making it unclear what $\text{MoM}(f,\mathbf{X})$ means: $\text{median}(\mu\_{f,x\_1},\ldots,\mu\_{f,x\_{\kappa}}).$ We will correct this in the next version of the paper, along with other suggestions from the reviewer to improve clarity.
>
> **Experiments:** As also mentioned by reviewer (giju), including experiments would be interesting, but we were pleased that neither reviewer required us to do so. Conducting such experiments would be quite involved. Testing the uniform convergence property would require running the estimation method (in this case, MoM) on all possible functions in the function class and then taking the maximum error. If the function class is infinite, this is infeasible. Thus, one would need to develop a reasonable discretization that is both computationally tractable and fine-grained enough to simulate uniform convergence over the class in the heavy-tailed case.
>
> We thank again the reviewer for their feedback and hope that we have addressed their concerns. If not, please let us know, and we will do our best to address them.

---

> > ### Comment · Reviewer_xSiA · 2025-04-08
> >
> > Thank you for your detailed response to my reviews. I appreciate that all of my concerns have been addressed. I will increase my overall ratings based on the discussions in the rebuttal section, provided that my concerns and suggestions are included in the next version.

---

### Official Review · Reviewer_giju · 2025-03-04

**Overall Recommendation:** 3

**Summary:**

The authors study the uniform convergence problem, i.e., estimating the mean of a set of functions simultaneously over inputs randomly sampled from an underlying distribution. Specifically, the authors focus on the classical median-of-means estimator that has celebrated performance on traditional distributional mean estimation, in particular in the heavy-tail regime. They show that median-of-means, under mild assumptions that the function class of interest can be "discretized" in some suitable sense, admits near-optimal sample complexity to achieve additive estimation error $\epsilon$ simultaneously over all functions in the class with probability $1-\delta$.

To prove their main theorem, the authors employ novel analytical techniques, including a novel symmetrization-discretization-permutation pipeline on the mean estimates from each bucket. They show that (1) for uniform mean estimation to fail, there must exist a function for which median-of-means has vastly different performances on a pair of symmetric inputs, (2) the function class can be discretized via a "relaxed $\epsilon$-net" argument to allow union bounds, even if the class is unbounded, and (3) for any fixed function, it is highly unlikely that mean estimate performs vastly differently on the symmetrization introduced in (1). The authors supplement their general theoretical findings with applications to clustering and linear regression, and raises questions of finding lower bounds to the problem setting.

**Claims And Evidence:**

The claims are believable and reasonable with supporting evidences. Median-of-means enjoys many folklore properties even beyond standard setting and heavy-tailed mean estimation, so I do not find the authors' conclusion surprising; It is however very nice to see a formal analysis of median-of-means under this setting.

While there are some details that can be illustrated better, the authors did also provided a relatively clear and intuitive proof to their results, and devises novel mechanisms for their analysis, which I believe should be appreciated.

**Essential References Not Discussed:**

To my knowledge the authors cite and discuss essential related literature comprehensively.

On a more peripheral aspect, relating to my aforementioned concern about the universal constants in Theorem 3.4: A line of work in finite-variance distributional mean estimation starting from Catoni (2012) culminates in the Lee and Valiant estimator (FOCS 2022), which achieves the optimal estimation error even up to constants. While it is unknown what the optimal constants are in heavy-tailed regimes, the Lee and Valiant estimator is believed to outperform and enjoy better constants than median-of-means as well. I wonder if there are evidences, theoretical or empirical, that rationalizes the magnitude of the universal constants the authors choose.

**Experimental Designs Or Analyses:**

This submission is focused on the theoretical analysis of median-of-means for uniform convergence mean estimation, and does not provide any experimental analysis.

I do not believe that any experimental setup is necessary, but it would be interesting to see experiments that compare the performance of empirical mean and other celebrated mean estimators to median-of-means on uniform convergence - if an appropriate experimental setup is available. For many mean-estimation related problems, especially in more general settings, the empirical mean often outperforms other sophisticated estimators that admit rigorous theoretical guarantees; I find it interesting to ask if this is the case in uniform convergence as well, or if the heavy-tailed regime calls for more sophistication than simply taking the empirical mean.

**Methods And Evaluation Criteria:**

The model chosen by the author, including the problem setting and the optimization over sample complexity, is standard for mean estimation and its derived problems. While the authors make assumptions on the discretizability of the input function class, they provide evidences that the assumption is mild and reasonable.

**Other Comments Or Suggestions:**

- While to my knowledge not enforced by the ICML submission style requirements, I believe it reasonable to format in-line citations such that the sentence is complete after removing the citations: for example, in Section 2, page 2, right column, instead of "...special case of the formulation given in (Oliveira & Resende, 2023) except...", use "...special case of the formulation given in Oliveira & Resende (2023) except...".
- In Section 3.2, page 3, there are multiple references to "Section 2" that appears to be referencing Figure 1 instead.
- Section 3.2, page 3, left column, line 160 1/2: Use `` instead of " for the left double quote.
- Section 3.4, page 5, left column, line 259 1/2: It is unclear immediately what does it mean for a random vector to be symmetric; A lot of other symmetric properties are not clearly defined in the paper as well.
- Section 3.4, page 5, right column, line 247 1/2: Extra comma in "...this discretization perserves, the imbalance..."

**Other Strengths And Weaknesses:**

The paper is overall relatively well-written and clear, but due to the technicality of their proofs and analyses, I believe some of the claims and definitions can be better motivated and intuited. As an example, in the analysis in Section 3.4 on page 5, the right column contains high-level explanations of the flow of the authors' argument in proving Theorem 3.4, with Lemma 3.6, 3.7, and 3.8, which in conjunction with Figure 1 on page 3, is very nice and appreciated, and paints a good high-level picture of the overall structure of their proof; the definitions of $\hat{\mathbf{S}}_\mathbf{b}^{(b)}(f, \epsilon)$ on the left column of page 5, however, are built upon many layers of prior definitions and are technically convoluted, without an immediately clear interpretation of what they are supposed to represent. A brief explanation (such as something along the line of "the fraction of the mean estimate with large/small deviation on different samples") may be necessarily helpful.

There are also some minor typos which I outline below.

**Questions For Authors:**

N/A.

**Relation To Broader Scientific Literature:**

Median-of-means is a well-celebrated and extensively studied mean estimator that enjoys many robustness properties in many extended regimes from classical finite-variance mean estimation, some of which folklore. The authors formally analyze its performance for the uniform convergence problem under heavy-tailed regimes, which to my and the authors' knowledge is the first such formal analysis. Previous works have studied other classical estimators under more limited or incomparable models, such as empirical mean under finite dimensional constraints, or trimmed mean with adversarial contaminations and fixed sample.

The authors' proposed novel techniques, including the alternative symmetrization of median-of-means and discretization of unbounded function classes, are potentially of independent interests for future work on mean estimation as well.

**Theoretical Claims:**

While I haven't examined technical content in the finest details, especially their choice of universal constants, the theoretical claims are reasonable to my best knowledge, and I do not identify any major issues with their proofs and analyses.

I am somewhat concerned with the universal constants in their sample complexity (Theorem 3.4), which seems to have quite large magnitude even in the standard finite-variance setting. A short discussion about the reasonableness of these constants (perhaps in comparison to the performance of empirical mean on settings for which it is shown to be optimal) will strengthen the authors' claims and arguments.

---

> ### Author Rebuttal · Authors · 2025-03-28
>
> We thank the reviewer for taking the time to read the manuscript carefully and provide feedback.
>
> **Theoretical claims:** We would like to address the reviewer’s comment about the magnitude of the constants in Theorem 3.4. We agree that the constants differ from the empirical mean estimate of a single function, and we provide the following explanation. The main reason for the larger constants is that we are trying to estimate multiple functions (possibly infinitely many), unlike in the case of a single function. In achieving this goal, the symmetrization, discretization, and permutation steps each introduce additional constant factors into the final bound, which leads to the larger constant. Furthermore, we did not optimize the constants in the proof of Theorem 3.4. We believe it is a good idea to add a comment about this in the next version of the paper and thank the reviewer for suggesting it.
>
> **Experiments:** We are pleased that the reviewer does not necessarily require us to include experiments in the paper, although we agree that it would be exciting to see the results of such experiments. However, conducting such experiments is quite involved. Testing for uniform convergence  would require running MOM and the empirical mean estimate on all possible functions in the function class and then taking the maximum of the error. If the function class is infinite, this is not feasible. Thus, one would need to develop a reasonable discretization that is both computationally tractable and sufficiently fine-grained to simulate uniform convergence over the class in the heavy-tailed case.
>
> **Relation to broader scientific literature:** We also make a brief remark on emphasizing that we are not the first to provide uniform convergence bounds under heavy-tailed noise, as described in the manuscript in the **Related Work** section. We differ from previous work due to the focus on the *sample complexity* of the problem instead that on the estimation error. Thanks to a new analysis technique, and due to the introduction of a novel complexity measure that, differently from previous work (focussing instead on the Rademacher complexity), is log of the size of the *relaxed* version of a discretization. Our main result (Theorem 3.4) enabled us to give improved sample complexity bounds in the important cases of k-means clustering and linear regression with general losses, which is instead unclear how to derive from previous work.
>
> **Relation with Lee and Valiant 2022:** We thank the reviewer for pointing out the breakthrough work from Lee and Valiant on optimal mean estimation. As noted above, while we didn't attempt to optimize the constants in our bounds, moving from the task of estimating a single mean to that of estimating the means of each function in a possibly infinite class is likely to introduce additional (and potentially large) constant factors for any estimator. On top of that, often, the estimates of *size* of the particular function class of interest suffer from possibly large constant factors that may hide the benefits even of asymptotically optimal estimators. As a result, proving bounds with optimal constants (which, at the best of our knowledge, are still unknown) in uniform convergence, is likely to require alternatives to the symmetrization, discretization, and permutations, in addition to the adoption of more refined estimators (e.g., Lee and Valiant 2022). This challenging problem is an interesting research direction for future work, and we will mention it in the next version of the manuscript.
>
> **Strengths and weaknesses:** We thank the reviewer for highlighting that we could have presented some of the more technical details in the proofs more intuitively and accessibly. For instance as pointed out of the reviewer, the definitions of $\hat{S}\_{b}^{>}(f,\varepsilon)$ and $\hat{S}\_{b}^{\leq}(f,\varepsilon)$ could be better explained, and we will do it in the revised version.
>
> **Other Comments or Suggestions:** We also thank the reviewer for providing several suggestions for improvement, which we will incorporate into the next version of the paper.
>
> If the reviewer has any further comments please let us know, and we will do our best to address them.

---

> > ### Comment · Reviewer_giju · 2025-04-06
> >
> > Thank you for your reply. Regarding my comment on constants in the "Theoretical claims" section, I meant to ask if the constants for sample mean on the uniform convergence problem is known in restricted settings, e.g., function classes with bounded interval and finite fat-shattering dimension, as outlined in the related works section.

---

> > > ### Author Response · Authors · 2025-04-06
> > >
> > > We thank the reviewer for engaging in the rebuttal.
> > >
> > > If we understand the comment correctly, the reviewer is asking for what order of magnitude constants are in uniform convergence bounds for the sample mean in well-known settings.
> > >
> > > We provide the following answer: In the canonical case of $[0,1]$-bounded functions the gap between the constants appearing in the upper and the lower bounds is large. Furthermore, the constants appearing in the upper bounds are big.
> > >
> > > In particular, for bounds based on the fat-shattering dimension of the function class, the multiplicative constant $C_1$ in the following *classical* upper bound from [P.L. Bartlett and P.M. Long 1996]
> > >
> > > $\left(\frac{C_1}{\varepsilon^2} \left(d\log^2\left(\frac{1}{\varepsilon}\right) + \log \left(\frac{1}{\delta}\right) \right) \right)$
> > >
> > > is at least $1536$, which can be seen from the proof of Theorem 9 (5) and taking $\alpha$ close to $\varepsilon/4$. This bound has recently been improved in [E. Esposito, R. Colomboni, A. Paudice 2025] to
> > >
> > > $\left(\frac{C_2}{\varepsilon^2} \left(d + \log \left(\frac{1}{\delta}\right) \right) \right)$
> > >
> > > where the constant $C_2$ is at least $5367c'$,  where $c'$ is supposedly a large unknown constant (see point (j) page 13).
> > >
> > > Finally, we also mention the specific case of binary valued functions, where the constant $c$ appearing in the upper bound of Theorem 1 of  [P.M. Long 1999] is at least $554$ (the estimate is obtained by Lemma 9 in the same paper).
> > >
> > > In terms of lower bounds, to our knowledge, the lower bound closest to the upper bound is that for uniform convergence over binary valued functions appearing in Section 28.2 (page 393) of [S. Shalev-Shwartz and S. Ben-David 2014], where the constant is at least $8$ (see the $m(\varepsilon,\delta) \geq (8d)/\varepsilon^2$ bound, line 5 in the beginning of Section 28.2).
> > >
> > > As one can see, the problem of establishing the optimal constants for uniform convergence is still an open problem, and the constants appearing in the upper bounds are in general large. We remark that especially the more complex cases, with fat-shattering dimension, which we believe we are closer to, suffer from large constants.
> > >
> > > We hope that this addresses the reviewer's comment, otherwise, we will be happy to further elaborate.
> > >
> > > **References:**
> > >
> > > [P.L. Bartlett and P.M. Long 1996]: More theorems about scale-sensitive dimensions and learning. Conference on Learning Theory (COLT). 1995.
> > >
> > > [P.M. Long 1999]: The Complexity of Learning According to Two Models of a Drifting Environment. Machine Learning. 1999.
> > >
> > > [S. Shalev-Shwartz and S. Ben-David 2014]: Understanding Machine Learning: From Theory to Algorithms. Cambridge University Press. 2014.
> > >
> > > [E. Esposito, R. Colomboni, A. Paudice]: An Improved Uniform Convergence Bound with Fat-Shattering Dimension. Information Processing Letters. 2025.

---

### Official Review · Reviewer_Libf · 2025-03-09

**Overall Recommendation:** 4

**Summary:**

The paper proposes to use the median-of-means estimator to (uniformly) estimate the mean over a whole real-valued function family, with respect to some unknown distribution with bounded $(1+p)$-th moment that we only get sample access from. The authors give an analysis of the maximum estimation error, under the assumption that the function family satisfies some approximability property with a small cover. They then applied the result to $k$-means clustering and to regression problems.

**Claims And Evidence:**

Yes.

**Essential References Not Discussed:**

No.

**Experimental Designs Or Analyses:**

N/A

**Methods And Evaluation Criteria:**

Yes.

**Other Comments Or Suggestions:**

N/A

**Other Strengths And Weaknesses:**

While the paper clearly *explains* the proof, I don't think it does as good a job with *motivating* the analysis and assumptions.

Let me walk the authors through what I was thinking when reading the paper, including an initial misunderstanding, the subsequent confusion and a partial resolution. Hopefully this can help improve the technical narrative in the paper.

1. Read the main sample complexity result. The form of the sample complexity looks easy to interpret -- it looks like a covering number term inside the log, so at this point I'm expecting a covering on the function class, and then a union bound net argument + approximation guarantees in between net elements.

2. I checked the assumption. The notion of D-discretization looks like what one would expect -- each $f$ gets mapped to some $\pi f$ in the net, so that $\pi f$ approximates $f$ well (Definition 3.1 has the sum of absolute differences over the samples being small, which is a strong condition). Then I was wondering, but why do we need the *three* sample sets? It clearly has something to do with the symmeterization argument that was foreshadowed, but why do we need that?

3. In fact, at this point I was wondering, why not do the obvious thing of taking a union bound over the net elements (with the net over the function class), and then use D-discretization to fill out the rest of the function class?

4. It took a while before I saw that the net in fact depends on the set of samples, which is why the simpler net argument fails. The D-discretization approximation condition is weaker in quantification than what one expects for a net argument. The key quantification difference is somewhat buried, and as a reader, it would be really helped if the difference was highlighted and emphasized.

5. But this still leaves the question, why was the weak notion of D-discretization used (which then seems to necessitate the complicated symmeterization), instead of the stronger "we have a single net that works for the entire function class (with high probability)" quantification? Is it because the latter stronger notion is impossible to prove for the applications at hand? If so, why, and shouldn't the applications then be the main point of the paper instead of just "median-of-means can be used for uniform mean estimation"?

=============

**Post-rebuttal discussion**: Thank you, this is exactly the sort of discussion I was looking for (as someone who hasn't paid too much attention to covering-based results), that prior works haven't found sample-independent covers. I have now raised my score, under the assumption that the authors will include this technical motivation and discussion of quantifiers in the paper.

**Questions For Authors:**

Please clarify the technical motivation as discussed above.

**Relation To Broader Scientific Literature:**

The uniform mean estimation problem is well-studied in the literature, both in the finite variance case (or bounded variance case, for Catoni-style estimators) and in the finite $1+p$-th moment case. Methods have been proposed, based on median of means, trimmed mean, and Catoni's estimator. This paper uses a symmeterization argument different from prior works.

A small personal gripe (though I don't actually reduce my score based on this) is the use of median-of-means. We know that the estimator is awful both theoretically and empirically, in the vanilla mean estimation problem (without any function classes). Both its finite sample performance and its (fix $\delta$, take $n \to \infty$) asymptotic performance are off by constants from optimal (definitely in the $1+p = 2$ case), which show up empirically. On the other hand, trimmed mean is at least asymptotically efficient, and the recent Lee and Valiant estimator is optimal in the constants both finite-sample and asymptotically. All of these other estimators are also minimax-up-to-constants optimal in "heavy-tailed" settings too, so it doesn't seem very good to still talk about median-of-means still, as clean and simple an idea it is.

**Theoretical Claims:**

I didn't check the proof details, but the high-level proof strategy does look like it should work. I have no correctness concerns.

---

> ### Author Rebuttal · Authors · 2025-03-28
>
> We first thank the reviewer for carefully reading the manuscript and providing constructive feedback.
>
> **Relation To Broader Scientific Literature:** We would like to comment on the reviewer's concern about the relevance of MoM in light of novel more refined estimators. We start noting that, similar to the MoM estimator, even the best-known bounds for the trimmed mean suffer from sub-optimal constants in the single mean task (see Oliveira, Ornstein, and Rico 2025, Theorem 1.1.1, for the case $p=2$). In addition, MoM is widely used as a sub-routine of other methods, including in the optimal Lee and Valiant estimator. Furthermore we notice that in the uniform convergence setting, differently from the single mean estimation case, it isn't even clear what the optimal constants are. Therefore, we believe that it still makes sense, to analyze the MoM in the more general uniform convergence setting. We thank the reviewer for eliciting such an interesting discussion.
>
> **Strengths and weaknesses:** We appreciate the reviewer’s efforts in walking us through their experience of reading the manuscript and highlighting that the current introduction of the $\mathcal{D}$-discretization is suboptimal. Regarding point 5), we would like to provide some high-level intuition about why we chose this weak notion of a net. As the reviewer pointed out, the discretization allows consideration of realizations of the samples $X_{0},X_{1},X_{2}\in (\mathcal{X}^{m})^{\kappa}$ to estimate any function $f\in \mathcal{F}$ except on a small fraction of the $\kappa$ subsamples $X_{0}^{i},X_{1}^{i},X_{2}^{i},$ where the approximation of $f$ can be arbitrarily bad. Since we are considering heavy-tailed distributions, we cannot expect that all subsamples will allow us to estimate the function accurately, thus why we did not see how to show the result with out this definition of the discretization allowing for the discretization to "fail" on a small number of subsamples. More specifically, in the case of k-means, we can assert that most of the mean estimates on the subsamples $X_{0}^{i},X_{1}^{i},X_{2}^{i},$ are small, but we cannot make any claims about the remaining mean estimates. With knowledge that the mean estimates on most subsamples are small, we can discretize the functions on these subsamples and disregard the discretization on the remaining ones. A similar argument is used in the proof of regression. As alloted to earlier we did not see a way to prove the theorem without this weaker notion of a net. We again thank the reviewer for emphasizing the need for a more detailed explanation of the discretization, which we will improve in the next version.
>
> If the reviewer has any additional comments, we would be happy to address them.

---

> > ### Comment · Reviewer_Libf · 2025-04-03
> >
> > I thank the authors for engaging in this discussion. I'm still a little bit confused though: I already understand that the index $i$ doesn't cover all of $[\kappa]$ (unsurprisingly for heavy tailed distributions we don't expect all the groups to concentrate well). My main question was rather, why is the cover $F_{(\epsilon,m)}$ chosen based on the samples $\mathbb{X}_0, \mathbb{X}_1, \mathbb{X}_2$, instead of fixed independently of the samples?
> >
> > If $F_{(\epsilon,m)}$ were hypothetically fixed independently of the samples, I think one could just argue that MoM estimates the functions in $F$ well with high probability by a union bound, and that the hypothetical Definition 3.1+3.2 imply that we can handle the rest of the function class from the net using the approximation? Am I missing something major? If not, then I believe this captures my main confusion: why does the cover $F$ need to depend on the samples $\mathbb{X}_0, \mathbb{X}_1, \mathbb{X}_2$, making the above simple argument fail?
> >
> > If the authors can explain this to me and incorporate this technical motivation into the paper, then I'd very gladly raise my score.

---

> > > ### Author Response · Authors · 2025-04-03
> > >
> > > We thank the reviewer for the engagement in the rebuttal.
> > >
> > > If we understood correctly, the reviewer is asking why the cover $F_{(\varepsilon, m)}$ is dependent on the realization of the sample, rather than being fixed beforehand, thus holding for all sample realizations.
> > >
> > > The answer is the following: it is possible to use a sample-independent cover, as long as you can find one since such a cover fulfills the requirements of Definition 3.1 and 3.2. As illustrated by sample-independent covers being captured in Definition 3.1 and 3.2, we see that the latter are more general notions leading thus to a more general result. The reason we went through these notions, is that canonical definitions of covering, go through sample-dependent covers (see for example Definition 27.1 in [S. Shalev-Shwartz, S. Ben-David 2014] where vectors should be thought as the values taken by the functions on the sample). Furthermore, to our knowledge, covering results guaranteeing the existence of a cover, are sample/distribution dependent: see for instance Lemma 7 in [A. Kupavskii and N. Zhivotovskiy 2020], where they state the bound in terms of packing (which yields a cover as well). We notice here that the cover is in terms of the uniform distribution over the given sample. For a further example, see Corollary 5.4 in [M. Rudelson and R. Vershynin 2006] where one has to take the distribution of the Corollary as the uniform over the sample to get a cover.
> > >
> > > Here one could also notice that, right after the symmetrization step, the empirical process of interest is (effectively) indexed by the projection of $\mathcal{F}$ onto the sample, which is arguably a simpler object compared to the one indexed by $\mathcal{F}$. This is also reflected in the definition of common complexity measures (including VC-/Pseudo-/Fat-Shattering-dimension) that indeed are related to the size of sample-dependent covers of $\mathcal{F}$.
> > >
> > > A related observation is that, since being sample-independent places stronger requirements on the cover, it is harder to find such covers. Indeed, beyond special cases (e.g., linear functions with bounded input and bounded weights), we are not aware of finite sample-independent covers for general function classes nor it is clear how to find them.
> > >
> > > Finally, we remark that these observations do not rule out the possibility of getting sample-independent covers, but highlight the fact that sample-dependent covers are the canonical approach to the discretization of $\mathcal{F}$, and as mentioned sample-independent covers are a special case of sample-dependent covers: thus this being the reason for the sample dependent notions of Definition 3.1 and 3.2.
> > >
> > > We hope that we where able to address the reviewer's comment else we would be happy to clarify further. We will include this discussion in the paper.
> > >
> > > **References:**
> > >
> > > [S. Shalev-Shwartz and S. Ben-David 2014]: *Understanding Machine Learning: From Theory to Algorithms.* Cambridge University Press. 2014.
> > >
> > > [A. Kupavskii and N. Zhivotovskiy 2020]: *When are epsilon-nets small?* Journal of Computer and System Sciences, 2020.
> > >
> > > [M. Rudelson and R. Vershynin 2006]: *Combinatorics of random processes and sections of convex bodies.* Annals of Mathematics. 2006.

---

### Official Review · Reviewer_18A3 · 2025-03-14

**Overall Recommendation:** 4

**Summary:**

In this paper, the authors tackle the problem of uniform mean estimation under heavy-tailed noise. Considering a set of functions, and a random variable, they analyze the sample complexity of providing a uniformly consistent estimation of the mean of the functions evaluated in the random variable.

**Claims And Evidence:**

All the claims are supported by proofs.

**Essential References Not Discussed:**

All the essential references have been discussed.

**Experimental Designs Or Analyses:**

There is no experimental campaign

**Methods And Evaluation Criteria:**

There is no experimental campaign.

**Other Comments Or Suggestions:**

See above.

**Other Strengths And Weaknesses:**

The paper presents the first algorithm tackling uniform mean estimation in the presence of heavy-tailed noise. The proofs seem correct and there is a nice technical work.

In some points the reading is not fluent, especially in the first sections, I would recommend stating the main results before giving an intuition of their proof. Moreover, I would suggest to "light" some constants to the closest one-point decimal value, at least. Just to make the text clearer.

**Questions For Authors:**

I have no relevant questions.

**Relation To Broader Scientific Literature:**

Existing literature deal with uniform mean estimation in the presence of non-heavy-tailed noise. This work advances the known results in the field in this sense.

**Theoretical Claims:**

I quickly went through the proofs. All of them seem correct and the results are reasonable

---

> ### Author Rebuttal · Authors · 2025-03-28
>
> We thank the reviewer for taking the time to read the manuscript and provide feedback.
>
> We also appreciate the reviewer’s suggestions for improving the organization and presentation by first giving the theorems and then the proof sketch.
>
> We also make a brief remark on emphasizing that we are not the first to provide uniform convergence bounds under heavy-tailed noise, as described in the manuscript in the **Related Work** section. We differ from previous work due to the focus on the *sample complexity* of the problem instead that on the estimation error. Thanks to a new analysis technique, and due to the introduction of a novel complexity measure that, differently from previous work (focusing instead on the Rademacher complexity), is log of the size of the *relaxed* version of a discretization. Our main result (Theorem 3.4) enabled us to give improved sample complexity bounds in the important cases of k-means clustering and linear regression with general losses, which is instead unclear how to derive from previous work.
>
> If the reviewer has any further comments, please let us know, and we will do our best to address them.

---

### Decision · Program_Chairs · 2025-05-01

**Decision:**

Accept (poster)

**Comment:**

The paper studies the problem of uniform estimation of the mean for a family of real-valued functions, under an an unknown distribution with bounded moments that we can sample. The main contribution is the analysis of the classical median-of-means estimator in thic context, under some standard assumptions on the underlying function family. Applications to clustering and regression are also given. Overall, the reviewers found the question to be interesting and the proof to be novel. Modulo some suggestion on the presentation of the technical results, the reviewers agreed that this paper is above the acceptance threshold.